# Gene therapy delivery of anti-Müllerian hormone in prepubertal female domestic cats induces long-term sterilization

Philippe Godin [1], Nicholas Nagykery[1], Natalie Sicher[1], Julie L. Barnes [2], Amy G. Miller[2], Christina Bunner[2], Amy K. Thompson [2], Motohiro Kano[1], Guangping Gao [3], Dan Wang [3], Patricia K. Donahoe[1], Linda Rhodes[4], David A. Brake[4], Thomas J. Conlon[4], William F. Swanson [2], Lindsey M. Vansandt [2] ✉ & David Pépin [1] ✉

The uncontrolled reproduction of free-roaming domestic cats exacerbates their welfare challenges and the ecological pressure they exert on wildlife populations. Because of logistic and economic constraints, surgical sterilization alone cannot scale to control the reproduction of the hundreds of millions of intact cats worldwide. Herein, we report that the single administration of an adeno-associated viral vector delivering an anti-Müllerian hormone transgene to prepubertal cats can fully prevent pregnancy once females reach adulthood. Treated kittens were closely monitored for up to 21 months to assess long-term health, transgene expression, reproductive hormones, and reproductive function. The intramuscular injection was well tolerated and did not impact physical growth. The sustained expression of anti-Müllerian hormone did not impact spermatogenesis in males. However, it induced sterility in mated females by preventing breeding-induced ovulation and increases in progesterone associated with luteal phases, resulting in safe and potentially lifetime sterilization in the female domestic cat.

The domestic cat (*Felis silvestris catus*) is a highly prolific species. Females reach puberty before 1 year of age, typically give birth to 3–5 kittens per litter, and can often raise two litters annually[1]. The uncontrolled reproduction of free-roaming domestic cats perpetuates stray and feral cat overpopulation. A subset of these unowned felines typically experiences a shortened lifespan without proper access to veterinary care and are more likely to be involved in traffic accidents or attacks[2]. Furthermore, predation by free-roaming domestic cats has been linked to unfavorable impacts on native species, particularly in ecologically vulnerable environments[3–5]. Currently, surgical sterilization is the most common method to control domestic cat populations, whether companion animals

sterilized by general practice veterinarians or unowned, free-roaming cats through *Trap-Neuter-Return* (TNR) programs. Numerous studies indicate that TNR programs yield effective results when applied intensively with adequate resources[6–9]. However, the scalability of TNR is often constrained by logistical, economic, and personnel-related challenges, as the surgical sterilization of cats requires trained veterinary professionals and substantial ongoing efforts[10,11]. Despite its demonstrated impact in specific contexts, broad-scale implementation of TNR for the estimated hundreds of millions of free-roaming cats worldwide remains unrealistic, and currently, no long-term alternative to surgical sterilization of female cats is approved by any regulatory body.

[1]Pediatric Surgical Research Laboratories, Massachusetts General Hospital, Department of Surgery, Harvard Medical School, Boston, MA, USA. [2]Center for Conservation and Research of Endangered Wildlife (CREW), Cincinnati Zoo and Botanical Garden, Cincinnati, OH, USA. [3]Horae Gene Therapy Center, University of Massachusetts Chan Medical School, Worcester, MA, USA. [4]Michelson Found Animals Foundation Inc., Los Angeles, CA, USA. ✉e-mail: lindsey.vansandt@cincinnatizoo.org; dpepin@mgh.harvard.edu

Vectored contraception is a strategy that employs the parenteral delivery of a nucleic acid-based transgene for long-term in vivo production of a target protein with contraceptive properties. The efficacy of this method was first demonstrated in female mice following the adeno-associated virus (AAV)-mediated delivery of transgenes encoding antibodies that bind key reproductive proteins[12]. Wild-type AAVs are small replication-defective, non-pathogenic members of the *Dependoparvovirus* genus that are composed of an icosahedral protein capsid containing a single-stranded DNA genome[13]. Recombinant AAVs (rAAVs) are engineered to encapsidate a transgene of interest along with the necessary regulatory elements. When delivered in vivo, they can sustain stable long-term transgene expression in a wide variety of tissues[13]. To date, all AAV-based vectored contraception approaches have used rAAVs to convert transduced tissues into biofactories that continuously secrete the contraceptive protein into the bloodstream, allowing its delivery to the gonads. Anti-Müllerian hormone (AMH), also known as Müllerian inhibiting substance (MIS), is a glycoprotein member of the transforming growth factor-beta (TGFβ) superfamily of ligands. In adult females, AMH is naturally produced by granulosa cells of growing follicles[14,15] and regulates multiple ovarian processes, including inhibition of primordial follicle activation and preantral follicle growth[16–19]. We have previously demonstrated that the AAV9-mediated delivery of an AMH transgene can suppress folliculogenesis and induce sterility in adult female mice[18]. AAV9 has a high tropism for skeletal muscle cells[20], which are long-lived and terminally differentiated, making it an ideal serotype for achieving a sustained expression of the therapeutic transgene. In adult female domestic cats, we have shown that AAV9 delivery of a domestic cat AMH transgene (termed *fcMISv2*) led to a multi-year inhibition of ovulation and completely prevented pregnancies[21], while others have reported an absence of births using a different AAV and AMH transgene[22]. Because of the delay between treatment and onset of sterility observed in mice (on the order of 4 weeks[18]), we speculate that treating female cats before sexual maturity would be advantageous to ensure therapeutic efficacy before the first estrus, and would be a potent tool in managing free-roaming cat populations. The safety and efficacy of this vectored sterilant have not been previously tested in prepubertal cats, and its effect on physical development associated with sexual maturation remains unknown. Furthermore, in adult males of most vertebrate classes, AMH is produced by Sertoli cells and has been implicated in the regulation of testosterone production by Leydig cells[23,24]. However, there are no comprehensive reports on the impact of elevated AMH on the health and fertility of male mice and domestic cats.

The objective of this study was to evaluate the safety and efficacy of an AAV9-fcMISv2 vectored sterilant injection in kittens of both sexes. Herein, we report no negative impact on health and physical development of cats of either sex receiving a prepubertal AAV9-fcMISv2 injection. This single injection did not impact spermatogenesis in males. In treated females, it allowed completion of puberty, with subsequent estrous cyclicity and mating activity, but fully abrogated ovulation and completely prevented pregnancy.

## Results

### Prepubertal injection of AAV9-fcMISv2 in domestic cats was well-tolerated and resulted in sustained elevation of serum AMH

To investigate the safety and sterilization efficacy of delivering a feline AMH transgene using AAVs to prepubertal male and female domestic cats, we recruited twelve kittens aged 2–3 months and weighing an average of 1.31 kg (1.07–1.73) at the time of injection (Fig. 1a and Supplementary Table 1). Kittens treated with the AAV9-fcMISv2 sterilant received either a low dose of $5 \times 10^{12}$ viral genomes per kilogram (vg/kg) (one male and four females) or a high dose of $1 \times 10^{13}$ vg/kg (one male and three females). Control animals (one male and two females) were injected with $5 \times 10^{12}$ empty AAV9 particles per kg (vp/kg) (Fig. 1A). Injections were performed intramuscularly in the right caudal

thigh under mild sedation. Injection site monitoring was performed daily for the first 14 days, weekly for up to 2 months, monthly until the end of the study (9- and 21-month post-injection for males and females, respectively), and yearly during post-study monitoring of females. We observed no redness, swelling, mass formation, or heat and pain at palpation. Kittens were housed in cages according to their treatment groups for the first 5 days and then transferred to a single-group enclosure.

To evaluate viral clearance, we assessed viral shedding in pooled fecal and urine samples collected daily for the first 5 days, and in individual oral swabs collected weekly for 2 weeks, which revealed low viral excretion in all groups (Supplementary Fig. 1). As expected, the average number of vector genome copies detected in the blood was significantly higher in AAV9-fcMISv2 treated cats than in controls ($P < 0.0001$ for both groups) (Fig. 1b). For all groups, the average circulating vector genome copies was significantly decreased 4 months after treatment when compared to the average levels before the 2-month time point ($P = 0.0044$ for controls and $P < 0.0001$ for both treated groups). These results suggest rapid viral clearance following treatment (on the order of days), followed by a slower turnover of infected short-lived cells (e.g., liver cells), which is largely resolved after 4 months.

To investigate the safety of the treatment, we evaluated a panel of serum hematology and chemistry parameters in treated cats. Serum amyloid A (SAA) is an acute phase protein (APP) and a reliable early indicator of systemic inflammation in cats[25,26]. SAA levels assessed during the first 2 months after the injection were not different between the control and the AAV9-fcMISv2 treated kittens and did not increase with time (Fig. 1c). No clinically significant changes were observed on physical examinations performed 3-, 6-, 9-, 12-, 18-, 26-, and 38-months post-injection. Complete hematologic and serum chemistry analyses were performed on blood collected at the same time points. One low-dose female (*Ahsoka*) had low red and white blood cell counts, combined with elevated concentrations of alanine aminotransferase (ALT) at the 12-month time point (Supplementary Fig. 2). However, these changes were likely secondary to sample clotting, as the physical examination was unremarkable, and all affected parameters returned within the normal range at 18 months. There were no clinically relevant changes in hematological markers or serum chemistry parameters associated with liver or kidney function (Supplementary Figs. 2 and 3). Of note, three cats (both treated males and one low-dose female, *Bellatrix*) showed a mild elevation in muscle enzyme creatine kinase concentrations throughout the study that was deemed clinically insignificant (Supplementary Fig. 3). Collectively, these findings demonstrate that the AAV9-fcMISv2 treatment was well tolerated.

Next, we sought to evaluate if the AMH protein expressed by the *fcMISv2* transgene triggered an immunogenic response in treated cats by measuring anti-AMH antibodies using a direct antigen-antibody capture ELISA. We did not detect any increase in AMH-binding IgG antibodies over background in the serum of kittens treated with AAV9-fcMISv2 at any of the time points assessed (Fig. 1d). The same assay successfully detected a 50-fold increase in AMH-specific IgG serum titers of a cat treated with a vector containing an immunogenic AMH transgene (*fcMISv1*) with sequence mismatches[21].

Similar to previous reports using the same AMH ELISA kit in female domestic cats[21,27,28], we found that physiological concentrations of AMH in control females ranged from 0.001 to 0.035 μg/mL throughout the study (Fig. 1e and Supplementary Fig. 4). Circulating AMH concentrations in females treated with either dose of AAV9-fcMISv2 increased steadily in the first 2 months, peaked between 10.8 and 48.2 μg/mL, then slowly decreased, and stabilized between 4.2 and 26.8 μg/mL at the 1-year post-injection time point (Fig. 1e and Supplementary Fig. 4). In contrast to females, mammalian males of prepubertal age have high physiological AMH concentrations that decrease as they go through puberty[29]. Baseline circulating AMH

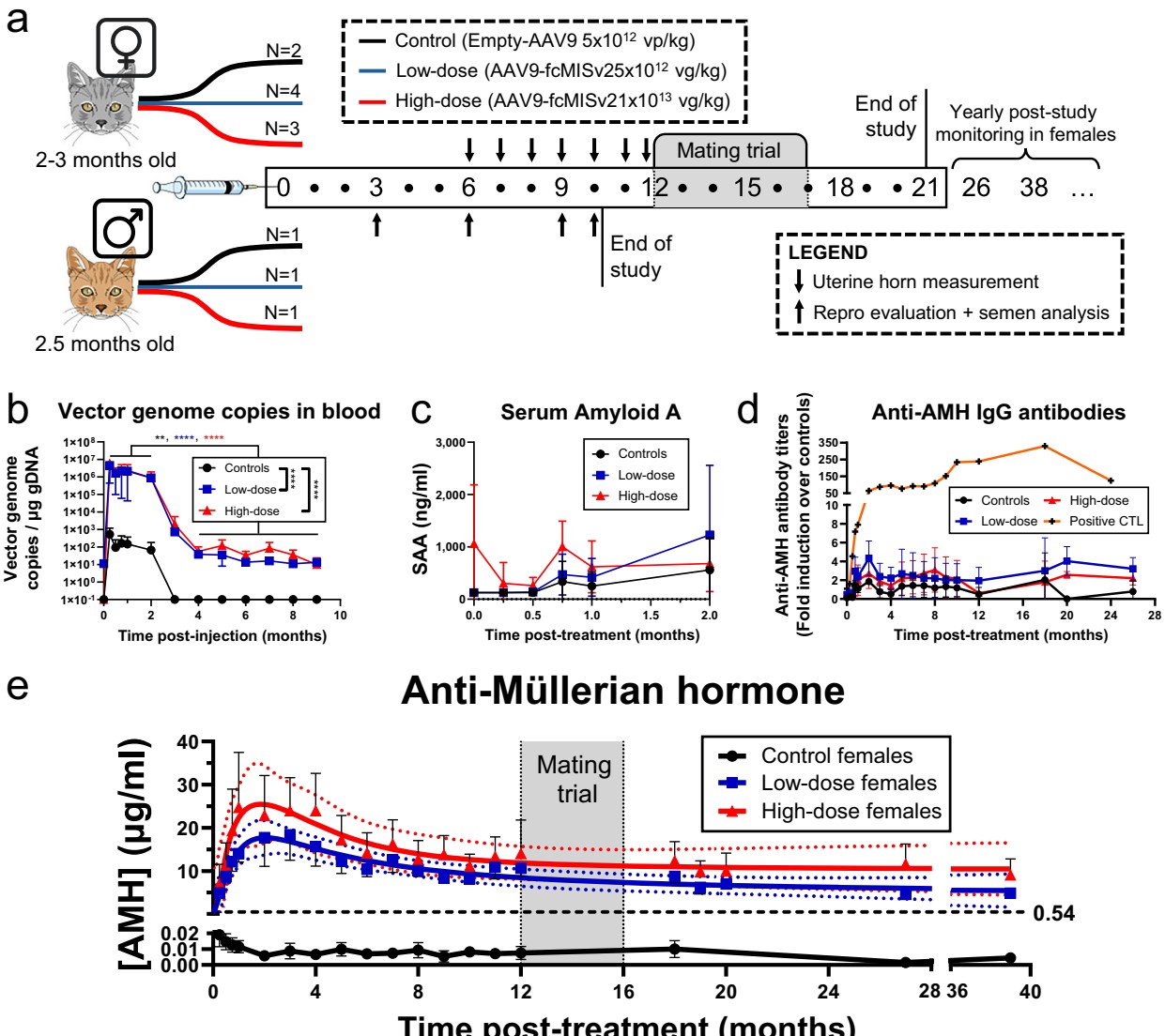

**Fig. 1 | The prepubertal injection of AAV9-fcMISv2 in domestic cats is safe and results in sustained elevated AMH concentrations. a** Nine female and three male kittens received a single intramuscular injection of either $5 \times 10^{12}$ viral particles per kilogram of body weight of AAV9-Empty vector (Controls, black), or $5 \times 10^{12}$ (Low-dose, blue) or $1 \times 10^{13}$ (High-dose, red) viral genomes per kilogram of body weight of AAV9-fcMISv2. The study lasted 21 months for females and included a single 4-month-long mating trial and ultrasound-guided uterine measurements (↓). Yearly post-study monitoring was performed at 26 and 38 months post-injection. The male study lasted 10 months and included four reproductive evaluations (↑). **b** Average vector genome copies (vg) per µg of genomic DNA (gDNA) quantified by qPCR in blood samples collected during the first 9 months post-injection. The average log-transformed vg/µg gDNA for cats in the low- and high-dose groups was compared to controls using a one-way ANOVA (Dunnett's post hoc test): ****$P < 0.0001$ for both. Within each group, the average vg/µg gDNA between 0.5 and 2 months was compared to the average after 4 months using a two-way ANOVA

(Šídák's post hoc test): **$P = 0.0044$ (controls) and ****$P < 0.0001$ (both treated groups). Datapoints with undetectable levels were plotted as 0.1. **c** Serum Amyloid A (SAA) concentrations during the first 2 months after the injection. **d** Anti-AMH antibody titers in the serum throughout the study. Serum samples of a cat injected with a vector containing an immunogenic AMH transgene are used as positive controls (orange, plus signs). **e** Circulating AMH concentrations in females throughout the study. A biphasic regression curve (solid line) was interpolated for both treated groups. The blue (low-dose) and red (high-dose) dotted lines indicate the 95% confidence interval. The horizontal dashed line indicates the lowest AMH reading (0.54 µg/mL) ever measured in an AAV9-fcMISv2-treated sterile adult female[21]. $n = 3$, 5, and 4 (**b**–**d**) and $n = 2$, 4, and 3 (**e**) cats in control (black circles), low-dose (blue squares), and high-dose (red triangles) groups, respectively. Data are shown as means ± SD (**b**–**d**) or means ± SEM (**e**). Cat and syringe illustrations were uploaded from the Mind the Graph platform (www.mindthegraph.com). Source data are provided as a Source data file.

concentrations in male kittens ranged from 0.78 to 1.18 µg/mL (Supplementary Fig. 4). While AMH concentrations decreased to 0.012 µg/mL in the control male 4 months later and remained low in adulthood, both AAV9-fcMISv2-treated males presented a pattern of elevated serum AMH concentration, with an early peak, slow decrease, and stabilization that paralleled our observations in treated females (Supplementary Fig. 4). Unexpectedly, serum AMH concentrations in AAV9-fcMISv2-treated kittens were generally higher than that of adults

treated with the same dose in a previous study[21], and remained well above the lowest AMH reading measured in a treated sterile adult female cat (0.54 µg/mL) (Fig. 1e). Together, these results indicate that the prepubertal injection of AAV9-fcMISv2 in domestic cats is well tolerated, and results in higher, sustained supraphysiological AMH concentrations than previously observed in adult cats[21], suggesting this timing may be advantageous for long-term efficacy and durability of this approach.

## Prepubertal treatment of female domestic cats with AAV9-fcMISv2 did not impair physical growth and estrogen production but impacted uterine size at maturity

To assess the effect of administering the AAV9-fcMISv2 sterilant before puberty on physical growth, we recorded body weights weekly throughout the study (Fig. 2a). Females in the high-dose group were significantly heavier than controls during the transition from the end of the juvenile growth phase to the beginning of the mating trial (between 8 and 14 months of age) ($3.34 \pm 0.23$ and $2.76 \pm 0.32$ kg, respectively; $P = 0.0296$). Females in the low-dose group had an average adult body weight ($2.88 \pm 0.11$ kg) comparable to controls during the same period. While adult body weight is greatly influenced by birth weight and food intake, all study animals' body weights fit between the 25th and 75th percentiles of a recently published growth standard chart[30], suggesting that their growth was normal.

We next sought to determine if supraphysiological AMH could impair puberty. Puberty is the transition period in the reproductive development of animals that is characterized by an increase in the production of gonadal steroids, the development of secondary sexual characteristics, and the manifestation of sexual behavior. Typically, female domestic cats become sexually mature between 8 and 12 months of age[1]. We measured estrogens as an indicator of gonadal steroid production. Overall, we were unable to detect a significant increase in fecal estrogen metabolites between the beginning of the study and the initiation of the mating trial in any group of females (Fig. 2b and Supplementary Fig. 5), suggesting estrous behavior may be a better indicator of puberty transition than fecal estrogen metabolites. Females in the low-dose group had a significantly higher average level of fecal estrogen metabolites before 8 months of age than controls ($701.8 \pm 405.6$ and $581.0 \pm 364.9$ ng/g of feces, respectively; $P = 0.0024$) (Fig. 2c). Females in the high-dose group had average fecal estrogen levels ($510.4 \pm 279.7$ ng/g of feces) comparable to controls during the same period. Overall, these results suggest that estrogen production is not physiologically impaired by supraphysiological AMH during puberty.

Because uterine physiology is influenced by sex steroids, and ovariectomy exerts a protective effect against uterine pathologies such as endometrial cystic hyperplasia[31], we evaluated if the uterus was affected by supraphysiological AMH. We measured uterine horn dimensions monthly by ultrasonography starting at 9 months of age. Since the average uterine cross-section area did not differ between groups at the 9-month time point, we can infer that supraphysiological AMH did not inhibit puberty-induced uterine growth. Unexpectedly, the average uterine cross-section areas between 9 and 14 months of age (the initiation of the mating trial) in treated females from the low- and high-dose groups were significantly smaller (4.3 and 4.7 cm², respectively) than that of control females (7.7 cm². $P = 0.0018$ and $P = 0.0030$, respectively) (Fig. 2d). These results suggest that supraphysiological concentrations of AMH reduce uterine diameter post-puberty, which may be indicative of a protective effect against hyperplasia. Collectively, these results indicate that the AAV9-fcMISv2 administration does not impair normal physical growth and estrogen secretion through puberty but is associated with a smaller uterus during early adulthood.

## Supraphysiological concentrations of AMH did not impair testicular development and function

Based on multiple reports suggesting AMH may inhibit *Cyp17a1* expression in Leydig cells and suppress testosterone production in rodents[23,24,32,33], we evaluated the impact of sustained supraphysiological concentrations of AMH on male fertility of mice and domestic cats. First, adult male mice received a single injection of $3 \times 10^{11}$ vg of AAV9 containing a human AMH transgene (AAV9-LR-hsMIS), a dose that produced complete sterilization in females[18]. Supraphysiological concentrations of AMH did not impact the male reproductive output (Supplementary Fig. 6a, b). Their serum testosterone concentrations were maintained over 1 ng/mL, and testicular weight was unaffected (Supplementary Fig. 6c, d). These results suggest that supraphysiological AMH concentrations in male mice did not disrupt androgen production and had no significant impact on their fertility.

Next, we sought to evaluate if the AAV9-fcMISv2 treatment was also benign in male domestic cats and if it could impact puberty timing. The approximate age of onset of hormonal puberty was defined herein as the age at which the average of six consecutive fecal testosterone metabolite readings was two standard deviations higher than baseline testosterone levels (Fig. 2e). This occurred at 6.3, 6.9, and 7.0 months of age in the control, low- and high-dose males, respectively. Figure 2f shows the average fecal androgen metabolite levels in males throughout the study ($2045 \pm 1319$, $1619 \pm 882$, and $2274 \pm 1904$ ng/g of feces for the control, low- and high-dose males, respectively). To further monitor sexual development, we performed a complete reproductive evaluation of the three males at 5, 8, 11, and 12 months of age. The average testicular volume of treated males increased after the age of five months, concomitant with the observed rise in testosterone production (Fig. 2g). In intact males, formation of the preputial cavity and eruption of the penile spines are sexual characteristics that develop secondary to androgen production and are indicative of sexual maturation. All three males in the study completed their normal penile development by 8 months of age (Supplementary Table 2). Histological analyses of testes collected at 12 months of age revealed an idiopathic oligospermia in the control male, preventing us from directly comparing control and treated testicular histology (Supplementary Fig. 7a). However, the seminiferous tubule diameter in the males treated with a low- and high-dose of AAV9-fcMISv2 ($258 \pm 18$ μm and $257 \pm 19$ μm, respectively) were consistent with the normal range reported in male cats[34] and were significantly higher than the (oligospermic) control male ($205 \pm 14$ μm. $P < 0.0001$ for both treated males) (Supplementary Fig. 7b). Inhibin B and luteinizing hormone (LH) concentrations assessed in monthly serum samples were similar between the three males (Supplementary Fig. 7c, d). These results suggest that supraphysiological AMH did not impair sexual development and puberty in male kittens.

To evaluate the fertility of AAV9-fcMISv2-treated male domestic cats, semen collection and analysis were performed as part of their reproductive evaluation (Supplementary Table 2). The ejaculate of all three males did not contain sperm at 5 months of age. Males treated prepubertally with AAV9-fcMISv2 presented expected semen volume and sperm concentration, progressive motility, and morphology at 8, 11, and 12 months of age, within the normal ranges expected in male cats[35–37]. The sperm of treated males at 11 months of age were assessed functionally with an in vitro fertilization trial using in vitro matured domestic cat oocytes. Their sperm showed sustained progressive motility and had the capacity to fertilize oocytes, as evidenced by the presence of blastomeric nuclei 48 h post-insemination (Supplementary Table 2). The idiopathic oligospermia observed in the control male precluded a parallel assessment of sperm motility, morphology, and function. Together, these results suggest that the administration of AAV9-fcMISv2 in prepubertal male domestic cats does not impair testicular development and function and likely does not prevent male fertility.

## Supraphysiological concentrations of AMH reduced estrogens and progestogens, increased LH secretion, disrupted estrous cycling, and prevented spontaneous ovulation

To characterize the impact of administering the AAV9-fcMISv2 sterilant prepubertally on ovarian physiology, we assessed the levels of key reproductive hormones in treated and control females. First, to evaluate the impact of supraphysiological AMH concentrations on the growing follicle pool, we compared the average circulating levels of inhibin B of treated females to the control group from 2 to 26 months post-injection (Fig. 3a). Females from the control and low-dose groups

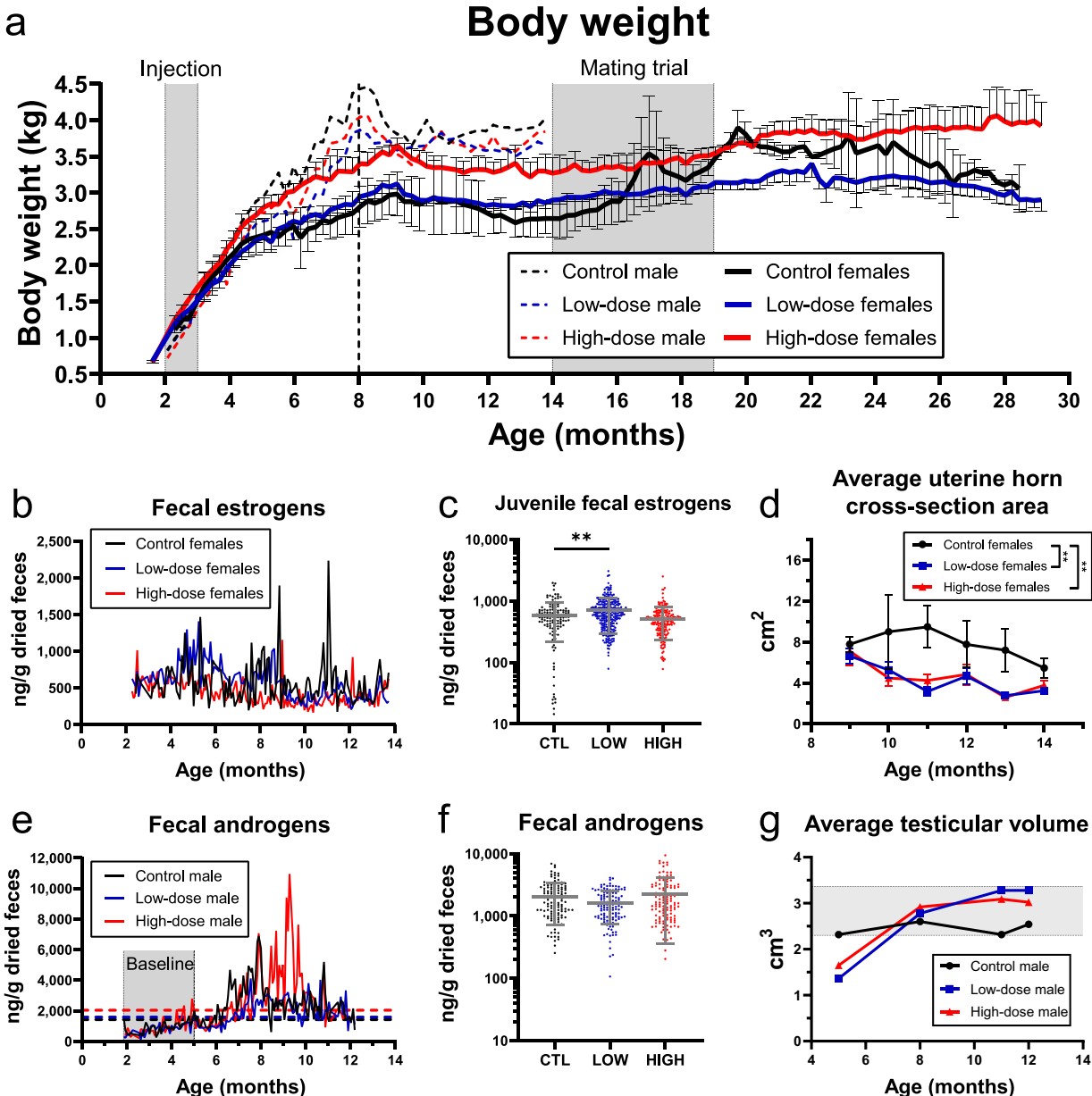

**Fig. 2 | The prepubertal injection of AAV9-fcMISv2 in domestic cats does not impair physical growth and testicular development but is associated with a smaller adult uterus. a** Individual (male, dotted lines) and average (female, solid lines) body weights of cats treated prepubertally with either $5 \times 10^{12}$ viral particles per kilogram of body weight of AAV9-Empty vector (Controls, black. $n = 1$ male, 2 females), or $5 \times 10^{12}$ (Low-dose, blue. $n = 1$ male, 4 females) or $1 \times 10^{13}$ (High-dose, red. $n = 1$ male, 3 females) viral genomes per kilogram of body weight of AAV9-fcMISv2. Shaded regions indicate the timing of treatment and the female mating trial period. The vertical dashed line indicates the approximate end of the juvenile growth phase. The body weights of females were plotted as means ± SEM. **b** Average fecal estrogen metabolite levels in females between the beginning of the study and the initiation of the mating trial. **c** Average fecal estrogen metabolite levels before 8 months of age. The average log-transformed data for females in the low- and high-dose groups was compared to controls using a one-way ANOVA (Dunnett's post hoc test): \*\*$P = 0.0024$ for the low-dose group. **d** Average uterine horn cross-section area of each group of females between 9 and 14 months of age. The average areas were compared to the control group using a one-way ANOVA (Dunnett's post hoc test): \*\*$P = 0.0018$ and 0.0030 for the low- and high-dose groups, respectively. **e** Individual fecal androgen levels in males throughout the study. The baseline fecal androgen level was calculated for each male based on the average levels between the ages of 2 and 5 months (shaded region). The horizontal dashed lines indicate the baseline + 2 standard deviations for each individual male. **f** Average fecal androgen metabolite levels in males throughout the study. **g** Average testicular volume in males at 4 different reproductive evaluations. The shaded region represents the average testicular volume ($2.83 \pm 0.53$ cm$^3$) in normozoospermic male domestic cats aged 12–24 months[64]. $n = 2, 4$, and 3 (**b**–**d**) and $n = 1, 1$, and 1 **e**–**g** cats in control (black, circles), low-dose (blue, squares), and high-dose (red, triangles) groups, respectively. **c**, **f** Repeated measurements collected 3 times per week are shown for each cat. Unless indicated otherwise, data are shown as means ± SD. Source data are provided as a Source data file.

had similar average circulating inhibin B concentrations during that period. However, females treated with a high dose of AAV9-fcMISv2 had significantly higher concentrations of inhibin B in the blood ($P = 0.0151$). Since inhibin B is lower during the luteal phase[38], we speculate that this finding reflected a suppression of luteal phases by supraphysiological AMH.

The female domestic cat is classified as an induced ovulator, with copulation serving as the canonical stimulus for ovulation. In absence

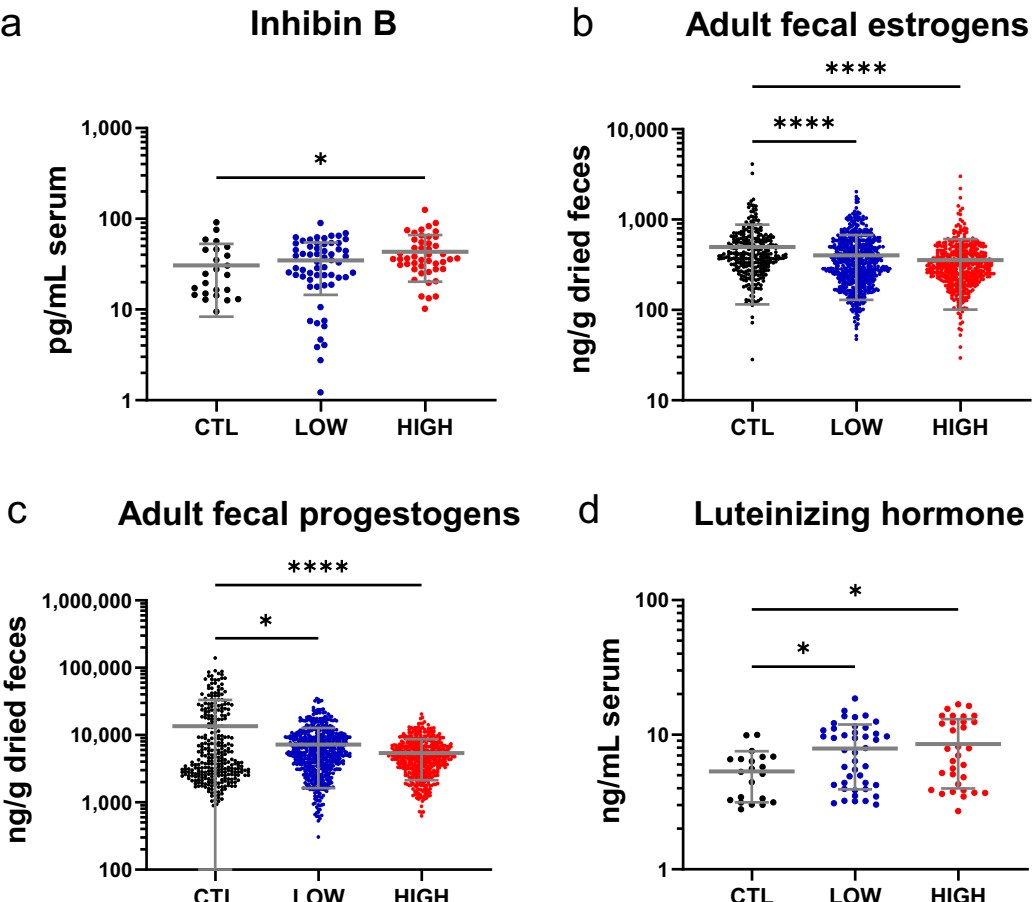

**Fig. 3 | Supraphysiological concentrations of AMH impair the reproductive hormone profile of female domestic cats.** Female domestic cats were treated prepubertally with either $5 \times 10^{12}$ viral particles per kilogram of body weight of AAV9-Empty vector (Controls – CTL, black) or $5 \times 10^{12}$ (Low-dose – LOW, blue) or $1 \times 10^{13}$ (High-dose – HIGH, red) viral genomes per kilogram of body weight of AAV9-fcMISv2. **a** Average circulating inhibin B concentrations in serum samples from 2 to 26 months post-treatment were compared to the average concentrations in the control group. Average estrogen **b** and progestogen **c** metabolite levels in fecal samples collected three times weekly after the age of 8 months until the end of the study in both groups of treated females were compared to the average levels in the control group. **d** Average circulating luteinizing hormone concentrations in serum samples collected monthly from 2 to 12 months post-treatment were compared to the average concentrations in the control group. Log-transformed data were compared using a one-way ANOVA (Dunnett's post-hoc test): *$P = 0.0151$ for inhibin B, ****$P < 0.0001$ for fecal estrogens and progestogens, *$P = 0.0144$ for fecal progestogens *$P = 0.0292$ (low-dose) and *$P = 0.0149$ (high-dose) for luteinizing hormone. $n = 2$, 4, and 3 female domestic cats in the control, low-dose, and high-dose groups, respectively. Repeated measurements collected monthly **a**, **d** or three times per week **b**, **c** are shown for each cat. Data are shown as means ± SD. Source data are provided as a Source data file.

of mating, certain females can also spontaneously ovulate[39,40]. In felids, steroid hormones such as estradiol and progesterone are excreted as metabolites primarily in feces and can serve as reliable indicators of ovarian cycle dynamics[41]. To evaluate the impact of the AAV9-fcMISv2 injection on subsequent adult estradiol levels, we compared the average levels of fecal estrogen metabolites between all three groups of females after 8 months of age, which is approximately when they reached their adult weight. Although we did not observe a reduction in estrogen metabolite excretion prepubertally (Fig. 2c), treated females from both groups excreted significantly lower average levels of estrogen metabolites than controls during adulthood ($497.3 \pm 381.2$, $401.9 \pm 272.2$, and $355.2 \pm 254.0$ ng/g of feces for the control, low- and high-dose groups, respectively; $P < 0.0001$ for both treated groups) (Fig. 3b). Comparison of the average levels of fecal progestogen metabolites during the same period revealed a similar effect. Female cats with supraphysiological AMH excreted significantly lower levels of progestogen metabolites than controls ($13,563 \pm 19,673$, $7,199 \pm 5,564$, and $5,395 \pm 3,257$ ng/g of feces for the control, low- and high-dose groups, respectively; $P = 0.0144$ and $P < 0.0001$ for the low- and high-dose groups, respectively) (Fig. 3c). This reduction corresponded to an

absence of peak progestogen metabolite readings (defined as values exceeding three standard deviations above baseline) in treated females; in contrast, control females had a bimodal distribution with high values corresponding to luteal phases. For our analysis, we defined a luteal phase as six or more consecutive peak progestogen metabolite measurements, which served as a proxy for ovulation status. Luteal phases were observed exclusively in control animals (Supplementary Fig. 5). Notably, the control female cat *Hermione* exhibited a spontaneous ovulation at 9 months of age; in contrast, spontaneous ovulations were never observed in any of the seven females that received a prepubertal AAV9-fcMISv2 treatment. Collectively, these results demonstrate that estrogen metabolites were moderately reduced, and progestogen peaks associated with luteal phases were abrogated with treatment, suggesting that supraphysiological AMH inhibits ovulation and normal cyclicity.

To further evaluate the impact of the AAV9-fcMISv2 injection on the hypothalamic-pituitary-gonadal axis, we compared the average circulating concentrations of LH in serum samples of treated females collected monthly from 2 to 12 months post-injection with samples from control females during the same period. Female cats from

low- and high-dose treatment groups had significantly higher concentrations of LH in the blood than controls (5.34 ± 2.14, 7.91 ± 3.93, and 8.51 ± 4.46 ng/mL for the control, low- and high-dose groups, respectively; $P = 0.0292$ and $P = 0.0149$ for the low- and high-dose groups, respectively) (Fig. 3d). We speculate that the observed increase in LH was in response to the reduced steroid production and absence of luteal phases in treated cats. Together, these results suggest that the prepubertal treatment with AAV9-fcMISv2 resulted in adult females that were acyclic, with reduced sex steroid secretions and no spontaneous ovulations. Treated females showed signs of mild hypergonadotropic hypogonadism with reduced estrogens and progestogens, and elevated LH, a phenotype similar to previous observations in treated mice[18].

### Prepubertal administration of AAV9-fcMISv2 prevented breeding-induced ovulation and pregnancy, and resulted in modified estrous behavior in adulthood

To assess the long-term sterilization potential of AAV9-fcMISv2 when administered before puberty we enrolled all females in a single 4-month-long mating trial that started a year after the injection. Cats were assigned to two mating groups (Supplementary Table 1). The two proven breeder males were introduced to their respective rooms 8 h per day, 5 days per week, and all male-female interactions were video-recorded for scoring breeding behavior. Breeding attempts were scored as successful if intromission and an appropriate post-copulatory behavioral response from the female were observed. Otherwise, it was scored as an unsuccessful breeding attempt (attempt to mount or mounting without intromission). All females underwent weekly transabdominal ultrasonography to monitor for pregnancy.

Importantly, both control females became pregnant, whereas none of the AAV9-fcMISv2-treated females had any confirmed pregnancies (Fig. 4a and Table 1). Surprisingly, while a majority (4/6) of treated adults in our previous study did not allow breeding[21], we observed breeding activity in all females treated as kittens, except one from the low-dose group. The six AAV9-fcMISv2-treated females that allowed breeding did so significantly more often than controls ($P = 0.0097$). On average, they displayed successful breeding attempts on 34.5% and 46.7% of the days spent with a male (low- and high-dose, respectively), while control females bred on only 14.7% of the days in the trial (Fig. 4b and Table 1). On days when successful breeding activity was recorded, the average number of successful attempts was similar between groups (Fig. 4c and Table 1). We defined a behavioral estrous period as consecutive days with at least one successful breeding attempt per day. The average length of behavioral estrus was similar in all groups (Table 1). However, AAV9-fcMISv2-treated females displayed a wider range of behavioral estrous length (Fig. 4d, e and Table 1). Additionally, single-day breeding was observed in five of the six treated females that bred. In comparison, all behavioral estrus observed in control females lasted between seven and 10 days. Phases in which treated cats showed estrous behavior were interspersed with interestrous phases of diverse lengths with multiple occurrences of very short duration (Fig. 4e and Table 1). These results suggest AAV9-fcMISv2 treatment increased the variability of estrous behavior timing and duration in cats and increased total mating interactions.

Following each ovulation, regardless of whether fertilization or implantation occurs, the queen enters a luteal phase characterized by elevated progesterone concentrations that suppresses further breeding activity[42]. Figure 4e shows a visual timeline of successful breeding activity and luteal phases in all females throughout the 4-month-long mating trial. Estrous periods in control animals were always followed by a luteal phase, while the variable estrous periods in AAV9-fcMISv2-treated females were not. The quantitative parallel assessment of behavioral and hormonal data in control females recapitulated the expected sustained increase in progestogen metabolite excretion that follows a breeding-induced ovulation (Fig. 4f). On the other hand, the

same analysis in females with supraphysiological AMH concentrations did not show any increase in fecal progestogen metabolites following an estrous phase. Together, these results indicate that female cats treated prepubertally with AAV9-fcMISv2 that bred did so more frequently and could exhibit a longer duration of breeding bouts than controls, likely because of the failure to induce ovulation that would normally terminate this behavior.

Overall, these data further support the hypothesis that supraphysiological AMH levels produced by the AAV9-fcMISv2 vectored sterilant preclude feline pregnancy by preventing breeding-induced ovulation and may modify estrous behavior by preventing luteal phases.

## Discussion

We have previously reported effective sterilization following the administration of a species-specific AMH transgene using AAV9-based vectors in adult female mice and domestic cats[18,21]. In the current study, we administered a single intramuscular injection of AAV9-fcMISv2 to prepubertal male and female cats and evaluated its impact on puberty, sexual hormones, and fertility. More than 3 years after treatment administration, circulating AMH concentrations remain above 1 μg/mL in all treated females. These levels were >200-fold higher than physiological concentrations reported in untreated adult female cats[21,27,43]. These levels were also higher than what we had previously observed when treating adult female cats with the same dosing and viral preparation, which suggests that prepubertal treatments may be preferable by inducing a higher production of AMH. We speculate that, given the efficient transduction of skeletal myocytes by AAV9 capsids[20], this increased production may be due to a combination of better transduction of young muscle cells and increases in muscle mass during puberty, which could translate into a more durable sterilization efficacy and/or the potential for reducing the treatment dose. To our knowledge, this is the first report of increased transgenic protein expression following the intramuscular administration of an AAV-based therapy to prepubertal animals compared to fully developed adults.

In the current study, abdominal ultrasonography was performed weekly in all females during the breeding phase to detect pregnancies. None of the seven females that received the vectored sterilant was ever declared pregnant. In contrast, a recently published study reported that 73% (8/11) of female domestic cats that received an AAV-AMH gene therapy with a modified AMH transgene conceived and had mid/late gestation fetal resorption[22]. Stocker and colleagues used a viral vector (AAVrh91) originally designed for intranasal airway infection in rhesus macaques[44] to deliver a feline AMH transgene with amino acid modifications to increase processing (i.e., a higher proportion of cleaved AMH), and AMH receptor binding activity. The reported decreased efficacy in pregnancy prevention despite otherwise similar treatment parameters raises the question of whether the AMH transgene modifications and/or difference in vector tropism may have reduced treatment efficacy. Importantly, our previous and current studies, along with the study by Stocker and colleagues, conclude that treatment with AAV-based vectors delivering feline AMH or AMH analogs prevents the birth of kittens in female cats, albeit through vastly different mechanisms.

In this study, none of the females treated prepubertally with AAV9-fcMISv2 showed any occurrence of breeding-induced ovulation, as evidenced by the absence of a sustained elevation in progestogen metabolites (i.e., a luteal phase) that is normally induced following a bout of steady breeding in this species. Spontaneous ovulation has been reported to occur in over 30% of domestic cats[39,40] and higher rates in group-housed females[45]. While we detected one spontaneous ovulation in a control female, we did not detect spontaneous ovulations in any of the seven treated females. This suggests that the sterilant may trigger an anovulatory phenotype by preventing both induced and spontaneous ovulations when administered prepubertally. We further evaluated reproductive hormones to

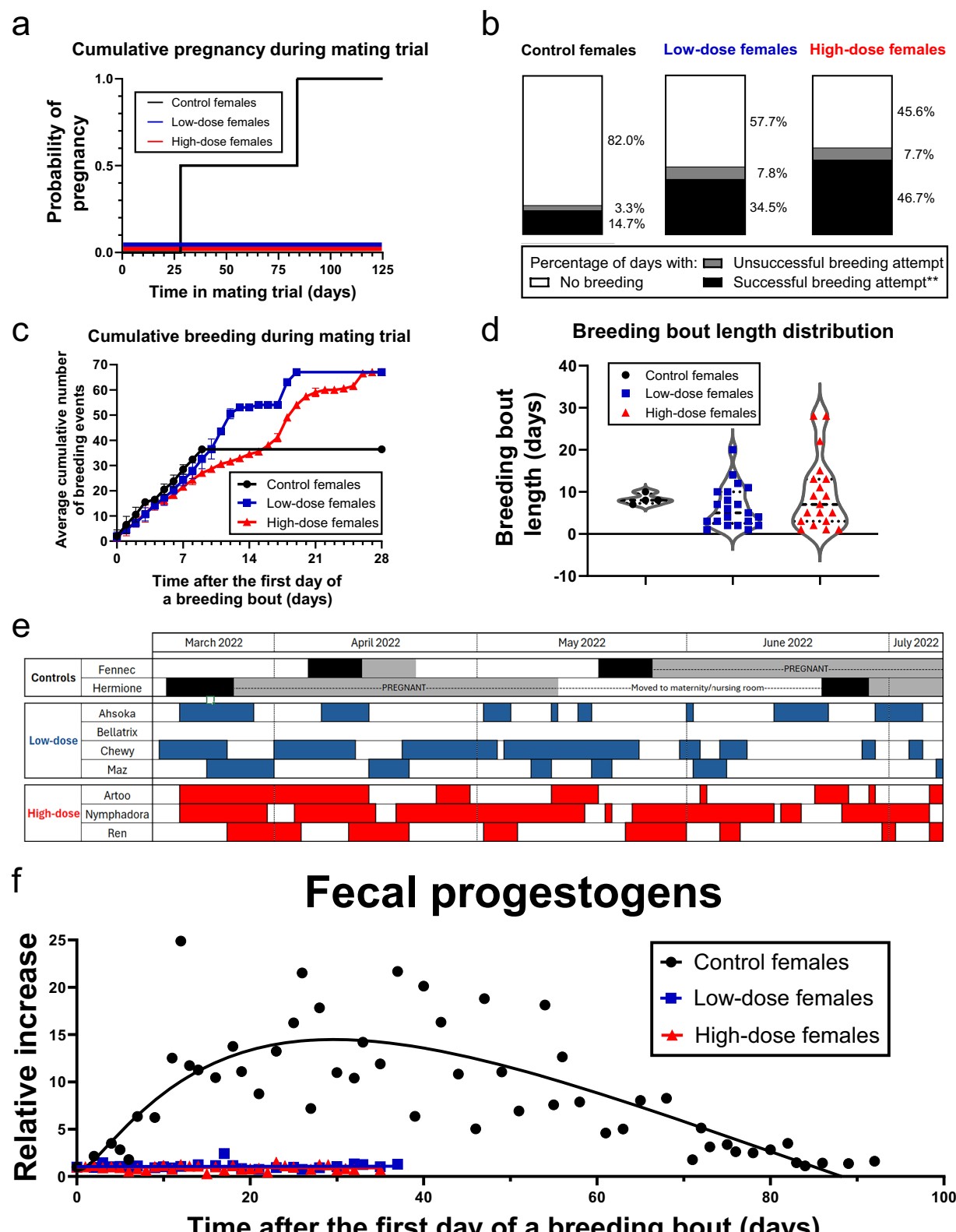

characterize the anovulatory phenotype in these cats. An inhibitory effect of AMH on aromatase expression and estradiol production has been previously suggested in multiple in vitro[46–48] and in vivo[49,50] studies. In this study, we observed evidence of moderate but statistically significant reductions in estrogen production in treated females compared to controls during adulthood. This contrasts with our previous findings suggesting supraphysiological AMH in adult female cats

did not significantly reduce the levels of estrogen metabolites when comparing pre- and post-treatment periods in individual cats[21]. Because our kittens were treated prepubertally, we were unable to compare pre- and post-treatment levels of estrogen metabolites. It is, therefore, possible that these differences were simply the consequence of inter-animal variability, particularly since this study only included two control animals to reduce unnecessary births.

**Fig. 4 | The prepubertal injection of domestic cats with AAV9-fcMISv2 prevents breeding-induced ovulation and pregnancy, and lead to disrupted estrous behavior.** Female domestic cats were treated prepubertally with either $5 \times 10^{12}$ viral particles per kilogram of body weight of AAV9-Empty vector (Controls, black, circles), or $5 \times 10^{12}$ (Low-dose, blue, squares) or $1 \times 10^{13}$ (High-dose, red, triangles) viral genomes per kilogram of body weight of AAV9-fcMISv2. **a** Inverted Kaplan-Meier survival plot showing the probability of pregnancy for each group during the mating trial. **b** Stacked histograms showing the average percentage of days that females from each group had at least one successful breeding attempt (black), no successful but at least one unsuccessful breeding attempt (gray) or no breeding interaction with a male (white). The percentage of days with successful breeding attempts was compared between the six treated females that bred and the two control females using an unpaired two-tailed T test with Welch's correction: \*\**P* = 0.0097. **c** Average cumulative successful breeding attempts for each group of females. Data are shown as means ± SD. **d** Violin plots showing the distribution of the duration of breeding bouts observed during the mating trial. Dashed lines indicate the median, dotted lines indicate the quartiles. **e** Timeline of successful breeding activity, luteal phases, and pregnancies. Days with at least one successful breeding attempt are shown in black, blue, or red. Luteal phases are shown in gray. Days with either only unsuccessful breeding attempts or no breeding activity and without elevated fecal progestogen readings are shown in white. **f** Average fecal progestogen metabolite readings relative to the number of days since the first day of the preceding breeding bout. Progestogen metabolite levels were normalized to the first day of a breeding bout for each individual cat. A bell-shaped, non-linear regression curve was interpolated for the control group (black line). A simple linear regression curve was interpolated for the low- (blue line) and high- (red line) dose groups. *n* = 2, 4, and 3 (**a**, **e**) and *n* = 2, 3, and 3 (**b**–**d**, **f**) female domestic cats in the control, low-dose, and high-dose groups, respectively. Source data are provided as a Source data file.

**Table 1 | Number of pregnancies, breeding activity frequency, and breeding bout details of female domestic cats during the mating trial**

|  | Control (n = 2) | Low-dose (n = 4) | High-dose (n = 3) |
|---|---|---|---|
| Number of pregnancies[a] | 2 | 0 | 0 |
| Total days in mating | 150 | 255[c] | 259 |
| Number of days with successful breeding attempts (% of total)[b] | 22 (14.7) | 88 (34.5)[c] | 121 (46.7) |
| Average number of successful breeding attempts per day (range) | 4.1 ± 2.4 (1–10) | 3.4 ± 2.4 (1–10)[c] | 2.7 ± 2.1 (1–12) |
| Average behavioral estrous period length in days (range) | 8.3 ± 1.1 (7–10) | 7.0 ± 4.7 (2–20)[c] | 11.1 ± 8.2 (2–28) |
| Average number of single-day breeding (range) | 0 ± 0 | 0.7 ± 0.9 (0–2)[c] | 1.0 ± 0.8 (0-2) |
| Average interestrous phase length in days (range) | N/A | 10.3 ± 7.0 (1–31)[c] | 8.3 ± 5.6 (1–21) |

[a]One-sided Fisher's exact test between controls (*n* = 2) and all treated females (*n* = 7). *P* = 0.0278.
[b]Unpaired two-tailed T test with Welch's correction between controls (*n* = 2) and treated females that bred (*n* = 6). *P* = 0.0097.
[c]Data are shown only for the three low-dose cats that allowed breeding.

Alternatively, these results could point to a reduction in antral follicle populations by supraphysiological AMH concentrations, as previously observed in female mice[18,21]. Importantly, levels of estrogen metabolites in treated animals remained within the normal range expected in cycling females[51,52]. We have previously demonstrated that supraphysiological AMH can inhibit the differentiation program of granulosa cells of murine preantral follicles, leading to reduced antral follicle numbers[19], and the steroidogenic differentiation of theca cells[53]. We speculate that a similar block in follicular development may occur in treated cats but is generally masked by endocrine homeostatic processes. Future studies, such as comprehensive histological assessments of the ovaries of AAV9-fcMISv2-treated females with follicle counts, along with single-cell transcriptomics, could provide a more detailed characterization of the impact of sustained supraphysiological AMH on domestic cat ovarian physiology and help elucidate the mechanism of anovulation. However, female domestic cats of this study were adopted intact to investigate their behavior in a home environment, precluding us from conducting such analyses. Considering the negative feedback of antral-follicle-derived estradiol on gonadotropin production by the pituitary, our findings of slightly reduced estrogens and progestogens and mildly elevated LH could be an indication of a mild hypergonadotropic hypogonadism phenotype. It remains unclear whether breeding-induced GnRH or LH release is disrupted in treated females, or if the response to LH is altered in the ovary, for example, due to a lack of antral follicles. Overall, our data is consistent with the notion that prepubertal administration of AAV9-fcMISv2 produces sterile adult females that cannot undergo breeding-induced ovulation and luteal phases. In contrast, this same approach in males does not appear to affect reproductive maturation or fertility. While several studies have previously reported a negative impact of AMH on androgen steroidogenesis in rodents[23,24,32,33], our results suggest that overexpression of AMH in male domestic cats did not impact

androgen production and its sequelae related to sexual development, sperm parameters and function, and fertility.

Single AAV9-fcMISv2 treatment completely suppressed breeding activity in four of six adult females[21]. Surprisingly, all but one treated female in this study (6/7) bred regularly, which confirmed that treated females had undergone puberty and reached sexual maturity. This allowed us to characterize the impact of the vectored sterilant on breeding patterns. In the current study, we speculate that the absence of luteal phases to terminate estrous behaviors in females treated with the AAV9-fcMISv2 sterilant was responsible for their extended receptivity to mating. Shille and colleagues[54] reported that the period of estrous behavior in queens lasts 7.4 ± 3.7 days on average, while the interestrous phase, which is characterized by the absence of mating behavior lasts 9.0 ± 7.6 days. The average duration of estrous and interestrous phases recorded in females treated with AAV9-fcMISv2 was similar. However, in our experience, the high proportion of estrous and interestrous phases that lasted less than 4 days (including single-day breeding events) that we observed in our treated females is atypical. We hypothesize that, in the absence of luteal phases and unambiguous periods of mating inhibition, the regulation of receptive and non-receptive periods normally flanking ovulation in females is disrupted by the AAV9-fcMISv2 treatment. We speculate that receptivity to mating, and even more so increases in mating frequency and duration, could translate into greater efficacy in the field for free-roaming cat population control over other methods (such as spaying) that take females out of the breeding pool, by diverting the attention of intact males away from fertile females.

However, it is unclear if the estrous behavior we observed in young treated females was an artifact of group housing in a laboratory setting with close contact with an intact male and cycling females, since these factors can promote estrous behavior[55]. It is possible that treated females in a household setting, not exposed to an intact male

or other cycling cats, may not be receptive to mating. Additional studies in a large number of AAV9-fcMISv2 treated cats living in home environments may be required to understand any effect of the treatment on estrous behavior.

Surgical sterilization, either via ovariectomy or ovariohysterectomy, is one of the most common surgical procedures performed in female cats worldwide. While this procedure is generally safe when performed by experienced veterinarians under proper surgical and anesthetic protocols[56], complications, such as intraoperative hemorrhage, postoperative wound swelling, infection or dehiscence, and self-inflicted incisional trauma, may occur[57]. Additionally, owners may be reluctant to choose surgical sterilization due to concerns about pre-existing medical conditions. In a field setting, this new approach would also require the capture of cats. While this mirrors a logistical requirement of TNR, administering an intramuscular product is considerably faster and less resource-intensive than surgical spay, requiring fewer personnel hours per animal. Under conditions where medical history is often unknown and access to medical supplies and trained veterinary personnel is limited, a non-surgical sterilant used as a complementary tool to TNR could prove to be highly valuable and offer many safety advantages. The close monitoring of the domestic cats involved in this study indicates that the AAV9-fcMISv2 female-specific sterilant is unlikely to cause adverse events, even if erroneously administered to male cats. Importantly, we did not observe any signs of acute or chronic systemic inflammation during the study timeframe. More than 3 years after prepubertal treatment with AAV9-fcMISv2, and over 5 years after postpubertal administration in our previous study[21], no injection site reactions have been observed in any of our study animals. Evaluation of hepatic, renal, and muscle serum chemistry parameters did not indicate clinically important abnormalities and the prepubertal administration of AAV9-fcMISv2 did not negatively impact weight gain and physical growth through puberty.

Given that uterine physiology is strongly influenced by sex steroids, and that ovariectomy may provide protection against uterine pathologies such as endometrial cystic hyperplasia and pyometra[31], we assessed uterine health using ultrasonography. Herein, we report reduced uterine horn diameters in females that received the AAV9-fcMISv2 sterilant. This could be secondary to a reduced estrogen stimulation because of estrogen's role in uterine epithelial cell proliferation[58]. A decreased endometrial and myometrial thickness has been reported in females in anestrus when compared to cycling queens[59]. Overall, the smaller uterine diameter in AAV9-fcMISv2-treated animals might indicate a protective effect of AMH in preventing endometrial cystic hyperplasia, perhaps by reducing frequency of ovulation and exposure to progesterone during the luteal phase, which may be evaluated in further studies.

Overall, the data presented herein support the excellent safety profile and extend the sterilization effectiveness of AAV9-fcMISv2 single administration from postpubertal[21] to prepubertal female domestic cats. Furthermore, this study suggests that prepubertal administration of AAV9-fcMISv2 may be preferable, by negating the impact of delayed onset of sterilization, and maximizing the elevation of circulating AMH concentrations to ensure sustained lifetime sterilization. This vectored sterilant approach is thus emerging as a promising, practical alternative to surgical sterilization of female domestic cats. While no gene therapy is commercially available for use in domestic animals, continual progress in the development and refinement of AAV technologies for human applications is paving the way for the implementation of AAV-based therapies in veterinary medicine, such as vectored sterilization of domestic cats.

## Methods
### Animals
Domestic cats (*Felis silvestris catus*) were maintained in a research colony at the Cincinnati Zoo and Botanical Garden's Center for Conservation and Research of Endangered Wildlife (CREW). All cat procedures were approved by the Cincinnati Zoo and Botanical Garden's Institutional Animal Care and Use Committee (Identification Number 22-170) and the Cincinnati Children's Hospital Medical Center Institutional Biosafety Committee (IBC 2021-0071). This study used female ($n = 9$) and male ($n = 3$) kittens aged 57-87 days at the time of treatment administration (Supplementary Table 1). Six female kittens were purchased commercially (Marshall BioResources), while the other kittens were born at the CREW facility. Kittens born in the colony were weaned at 7-8 weeks of age, while commercially purchased kittens were weaned shortly before their arrival at 5-6 weeks of age. Kittens were housed under a 14:10 h light:dark cycle, fed Purina Pro Plan Development Chicken and Rice kitten dry food (Nestle Purina Petcare), and provided access to fresh water ad libitum throughout the study. Food was switched to Purina Pro Plan Complete Essentials Chicken and Rice adult dry ration (Nestle Purina Petcare) when study animals were approximately 1 year old. Kittens were group housed under Biosafety Level 2 containment according to their treatment group for 5 days after vector administration and then returned to a single colony room for group housing. Male kittens were moved to individual enclosures 2 months after treatment administration to prevent potential breeding and fighting, while females were group-housed in a single room until the mating trial. The two proven male breeders (2.3 and 2.8 years old at the initiation of the mating trial) were singly housed outside of breeding periods. All cats except male breeders were made available for adoption as companion animals at the end of the study.

Mice experiments were conducted with 8-weeks-old C57BL/6 male mice ($n = 5$) (Charles River Laboratories) and were performed in accordance with the experimental protocol 2014N000275 approved by the Massachusetts General Hospital Institutional Animal Care and Use Committee. The mice were housed under a 12:12 h light:dark cycle, with room temperature maintained at 20-23 °C and humidity levels kept between 30 and 70%. Mice had unrestricted access to water and to Prolab IsoPro RMH 3000 (LabDiet—cat#0006972) rodent chow. Mice were humanely killed using either $CO_2$ asphyxiation and cervical dislocation, or by intraperitoneal barbiturate overdose (100 mg/kg pentobarbital, Vedco) after intraperitoneal ketamine (100 mg/kg, Dechra)-xylazine (10 mg/kg, Covetrus) anesthesia.

### Transgene design and AAV vector manufacturing
We engineered the *fcMISv2* transgene using the domestic cat AMH NCBI reference sequence *XP_011286375.2*, and optimized codon usage to reduce GC content using the GenSmart Codon Optimization algorithm (GenScript). The *LR-hsMIS* transgene differs from the wild-type human AMH gene by the substitution of its endogenous leader peptide sequence (L) with that of human albumin and the Q425R substitution (R) at the cleavage recognition site to enhance proteolytic activation[60]. The gene expression cassette in both recombinant AAV (rAAV) expression plasmids consists of a cytomegalovirus (CMV) enhancer with chicken β-actin promoter (CB6) driving the transgene, followed by a rabbit β-globin polyadenylation signal. The entire gene expression cassette is flanked by AAV2 inverted terminal repeats serving as packaging signals. AAV vectors were produced by a triple-transfection method in HEK293 cells, purified by cesium chloride sedimentation and dialysis[61]. Titers were determined by droplet digital PCR using a Taqman reagent assay (Fwd: 5'-GCCAAAAATTATGGGGACAT-3', Rev: 5'-ATTCCAACACACTATTGCAATG-3', Probe: 6FAM-ATGAAGCCCCTT-GAGCATCTGACTTCT-TAMRA) (for rAAV genome) and gel electrophoresis followed by silver staining (for rAAV capsid). The final vector formulation consisted of 5% sorbitol in phosphate-buffered saline (PBS), pH 7, with 0.001% Pluronic F-68 (MilliporeSigma).

### Treatment administration
The twelve study kittens were assigned to one of three groups: (1) Control (2 females (F)-1 male (M): $5 \times 10^{12}$ viral particles per kg of body

weight [vp/kg] of AAV9-Empty), (2) Low-dose (4F-1M: $5 \times 10^{12}$ viral genomes per kg [vg/kg] of AAV9-fcMISv2) or 3) High-dose (3F-1M: $1 \times 10^{13}$ vg/kg of AAV9-fcMISv2). Kittens were randomly assigned using the random (RAND) function in Microsoft Excel while ensuring that each group contained one of the three females born at CREW. Supplementary Table 1 lists group assignments and treatment administration information. On Day 0 of the study, prior to treatment administration, kittens were anesthetized using a single intramuscular injection of ketamine (1 mg/kg, Covetrus), dexmedetomidine (0.01 mg/kg, Zoetis), and butorphanol (0.2 mg/kg, Zoetis) in the left thigh muscle. After shaving and disinfection of the injection site, the control and treatment injections were performed in the caudal right thigh muscle with a maximum volume of one mL per site. Anesthesia was partially reversed with an intramuscular injection of atipamezole (0.1 mg/kg, Zoetis).

Adult male mice received a single intraperitoneal injection of $3 \times 10^{11}$ vg of either AAV9-GFP ($n = 2$) or AAV9-LR-hsMIS ($n = 3$) per mouse (equivalent to approximately $1.5 \times 10^{13}$ vg/kg).

### Health monitoring
Safety monitoring included daily assessment of general health during the whole study (10 and 21 months for males and females, respectively). Periodic evaluation of injection sites was performed daily for 14 days, weekly for 2 months, monthly until the end of the study, and yearly in females during post-study monitoring. Physical examinations and blood work (HealthChek™ Profile, IDEXX Laboratories) were performed prior to study enrollment, and at 3-, 6-, 9-, 12-, 18-, 26-, and 38-months post-injection, with the 12-month and subsequent timepoints conducted only in females. Cats were weighed weekly throughout the study, beginning 3 weeks prior to treatment administration.

### Sample collection
To quantify the shedding pattern of AAV9-fcMISv2 in domestic cats, urine and fecal samples were collected from each kitten during the first 6 days post-injection and were pooled into a single daily sample per group. Individual saliva samples were collected with sterile collection swabs (Zymo Research) at Days 0 (pre-treatment), 2 (males only), 7, and 14. Whole venous blood was collected in microtainer EDTA tubes at Days 0 (pre-treatment), 7, 14, 21, 28, and monthly thereafter through Month 9 to assess viral persistence in circulation. All samples were stored at −20 °C until analysis. Venous blood samples collected in sample separator tubes (Avantik) at Days 0 (pre-treatment), 7, 14, 21, 28, monthly through Month 21 (except during the mating trial), and at Months 26 and 38 were allowed to clot for fifteen minutes and then centrifuged for 10 min at $2000 \times g$. The serum samples were transferred into cryovials and stored at −80 °C until analysis. To facilitate fecal sample identification by individual cat, cats were fed small amounts of Purina Pro Plan Development Chicken and Liver or Complete Essentials Beef and Carrots Entrée kitten and adult wet food (Nestlé Purina PetCare) containing food-grade dye (Wilton Brands) and/or glitter (Dixon Ticonderoga Company) on the night preceding sample collection. Fecal samples were collected on 3 non-consecutive days per week to quantify estrogen, progestogen, or androgen metabolites beginning 2 weeks prior to treatment and concluding ten (males) or seventeen (females) months after treatment. Fecal samples were sealed in plastic bags and stored at −20 °C until processing.

Mouse blood collected by facial vein puncture was left to clot for 30 min in an Eppendorf tube and then centrifuged for 15 min at $2000 \times g$. Drawn serum was transferred immediately to a new tube and kept frozen at −20 °C until analysis for AMH and testosterone. Testes and epididymides were collected and weighed together at the end of the mating trial.

### Quantification of viral genome excretion and persistence
Quantification of target vector genome copies (vg) in urine, feces, saliva, and blood was performed by the Powell Gene Therapy Center (University of Florida). Briefly, genomic DNA (gDNA) was isolated from oral swabs and blood samples using the DNeasy Blood and Tissue kit, from urine using the QIAamp Viral RNA kit, and from feces using the QIAamp Fast DNA Stool kit (all from Qiagen) following the manufacturer's instructions. DNA concentrations were determined using the Quant-iT PicoGreen dsDNA Assay kit (Invitrogen). Target vgc present in gDNA were quantified by real-time PCR using the QuantStudio 3 real-time PCR system (Thermo Fisher Scientific) and analyzed using the QuantStudio Design and Analysis version 1.4.1 software (Thermo Fisher Scientific). Primers (Fwd: 5′-CATCTACGTATTAGTCATCGCTATTACCA-3′; Rev: 5′-CCCATCGCTGCACAAAATAATTA-3′) and probe (6FAM-CCACGTTCTGCTTCACTCTCCCCATC-TAMRA) were designed to the CB promoter sequence included in the AAV9-fcMISv2 vector design. PCR reactions were performed in triplicate, contained a total volume of 50 μL, and were run at the following conditions: 50 °C for 2 min, 95 °C for 10 min, and 45 cycles of 95 °C for 15 s and 60 °C for 1 min. Cycle threshold (Ct) values for each unknown sample were plotted against a standard curve using plasmid DNA containing the CB promoter sequence, and data were reported as vg/μg gDNA.

### Ultrasonography (for uterine horn measurement and pregnancy detection)
Uterine horn measurements were performed by transabdominal ultrasound examinations monthly starting at 9 months of age. Measurements were taken on cross-section view of each uterine horn, at their closest point to the uterine body. The widest (D) and narrowest (d) diameters were measured for each horn, and the area was calculated as an ellipse (area = π × 0.5D × 0.5d). The average uterine horn cross-section area (cm²) is reported as the mean between the left and right horns. Abdominal ultrasonography was performed weekly during mating trials for the detection of pregnancy. Pregnant females were re-assessed weekly to monitor fetal development, evaluated radiographically at 7–8 weeks post-breeding to determine fetal number, and then transferred to isolated maternity cages approximately 2 weeks prior to parturition to provide an appropriate birth setting. For ultrasonography procedures, females were placed in dorsal recumbency, and a 3–11 MHz microconvex and/or 4–15 MHz linear transducer (Esaote S.p.A.) and MyLab OMEGA ultrasound system (Esaote S.p.A.) were used.

### Enzyme immunoassays
Serum Amyloid A (SAA) circulating concentrations were measured in all cat serum samples drawn during the first 2 months of the study using the Feline Serum Amyloid A Detection ELISA Kit (Chondrex). All serum samples were diluted 1:2 or 1:4 and assayed following the manufacturer's instructions. Readings that were under the standard curve were plotted at 31.25 ng/mL (lowest standard) multiplied by their dilution.

We previously designed and optimized a direct antigen-antibody capture ELISA to quantify AMH-binding IgG antibodies in cat serum[21]. Briefly, wells were coated with recombinant FLAG-tagged feline AMH protein produced by CHO-K1 cells. Standard wells were coated with eight dilutions of cat IgG (Rockland Immunochemicals−cat#002-0102-0002. Lot#32041) in coating buffer. The plate was then blocked with bovine serum albumin (Jackson ImmunoResearch) and normal goat serum (Abcam Limited) in PBS containing 0.1% Tween-20 (Bio-Rad Laboratories) (PBST). Monthly serum samples were diluted 1:100 in blocking buffer, added to the plate, and incubated for 1 h at RT. The plate was then incubated with polyclonal goat anti-feline IgG (H + L) horseradish peroxidase (HRP) secondary antibody (Novus Biologicals−cat#NBP1-73347. Lot#4168) 1:10,000 in PBST for 1 h, and the 3,3′,5,5′-Tetramethylbenzidine (TMB) (MilliporeSigma) substrate reaction was performed. The reaction was stopped with the addition of 2 N sulfuric acid, and light absorbance was read at 450 nm.

The AMH Gen II ELISA kit (Beckman Coulter) was used for the quantification of circulating AMH concentrations in cats following the manufacturer's instructions. Serum was diluted in sample diluent as follows: control female samples for all time points and baseline treated female samples, 1:4; control male samples collected 3 months post-injection and at subsequent time points, 1:10; baseline males and control male until 2 months post-injection, 1:200; *Crabbe*, *Ahsoka*, *Bellatrix*, *Maz* and *Nymphadora* samples collected 7 days post-injection and at subsequent time points, 1:1000 to 1:2000; *Chewy*, *Goyle*, *Artoo* and *Ren* samples collected 7 days post-injection and at subsequent time points, 1:3000 to 1:5000.

Human AMH in mouse serum was measured using an in-house ELISA[60]. Briefly, a 96-well flat-bottom Immulon 2 HB ELISA plate (Thermo Fisher Scientific) was coated with 5 μg/mL mouse monoclonal anti-human recombinant AMH antibody (clone 6E11) overnight at 4 °C and was blocked with bovine serum albumin in PBST for 2 h at RT. After washing in PBST, 50 μL of the AMH standards or mouse serum samples diluted up to 1:500 was added to the wells and incubated overnight at 4 °C. A rabbit polyclonal anti-AMH antibody (MGH6) was then added at 1:4000 dilution and incubated 1 h at RT. HRP-conjugated donkey anti-rabbit IgG polyclonal antibody (Jackson ImmunoResearch—cat#711-035-152) was added at 1:35,000 and incubated 1 h at 4 °C. The plate was developed using TMB, the reaction was stopped with sulfuric acid, and light absorbance was read at 450 nm.

Fecal samples assessed for reproductive hormone metabolites were lyophilized, pulverized, and extracted by mixing with 90% ethanol at a 1:10 (w:v) ratio overnight[62]. To determine estrogen metabolites, a polyclonal antibody produced against 17β-estradiol (R4972, Coralie J. Munro—University of California) was used at 1:10,000 in conjunction with an HRP-conjugated ligand[62]. To measure progestogen and androgen metabolites, commercial assays (Progesterone ISWE Mini-Kit and Testosterone ISWE Mini-kit, Arbor Assays) were used with extracted fecal samples following the manufacturer's instructions.

Circulating inhibin B concentrations were measured in undiluted monthly serum samples using the Cat Inhibin B (INHB) ELISA kit (MyBioSource) and following the manufacturer's instructions.

Luteinizing hormone (LH) concentrations were analyzed by the Wildlife Endocrinology Research Laboratory (Smithsonian's National Zoo and Conservation Biology Institute) in diluted serum samples (1:2 to 1:10) using a double enzyme immunoassay adapted from[63]. Briefly, after coating 96-well plates with anti-mouse IgG (Arbor Assays) and overnight blocking, 50 μL of NIH-bovine LH standards, controls, or samples were added to wells. Monoclonal mouse anti-bovine LHβ antibody (clone 518-B7, Janet F. Roser—University of California) was added to all wells at 1:400,000. After overnight incubation, biotinylated ovine LH (NIDDK-oLH-26, National Institutes of Health—National Hormone and Pituitary Program) was added to all wells at 1:200,000 and allowed to compete for 4 h at RT. Plates were then incubated with streptavidin-peroxidase conjugate (F. Hoffmann-La Roche), followed by TMB substrate solution (Thermo Fisher Scientific), and reaction was then stopped using 1 N HCl.

Murine testosterone serum concentrations were measured with a testosterone ELISA (Immuno-Biological Laboratories) by the Ligand Assay and Analysis Core of the Center for Research in Reproduction (University of Virginia) following the manufacturer's instructions.

For all assays, standards, and samples were analyzed in duplicate. For SAA, AMH-binding antibodies, AMH, Inhibin B, and testosterone ELISAs, luminescence was detected using the Wallac 1420 Victor2 microplate reader (PerkinElmer), data were collected using the Wallac 1420 version 3.00 software (PerkinElmer), and quantitative analysis was performed using the "Four Parameter Logistic Curve" online data analysis tool (MyAssays Ltd.) accessible at https://www.myassays.com/four-parameter-logistic-curve.assay.

## Mating trials

A single 4-month-long feline mating trial was initiated approximately 1 year after treatment (Supplementary Table 1). Females were divided into two separate mating groups because three cats were direct descendants of one of the two breeder males. The week before initiating the mating trial, males were introduced to their respective rooms through a 5-day-long controlled exposure where the cats could see and smell each other for 2 h per day without any possibility of mating. Once the mating trial started, males were group-housed in their respective rooms for 8 h per day, 5 days per week, and continually video-recorded for analysis of breeding behavior. A breeding interaction was scored as a successful breeding attempt if intromission and an appropriate post-coital behavioral response from the female (rolling and/or licking of the vulva) were observed. Failure to mount or mounting without intromission was scored as an unsuccessful breeding attempt. A behavioral estrous period was defined as consecutive days with at least one successful breeding attempt per day. An interestrous phase was defined as the interval between two successive estrous periods if no luteal phase was detected. Pregnant females confirmed via ultrasonography were relocated to single-occupancy, double-compartment maternity housing around day 50 of gestation.

Adult male mice were housed with untreated adult females (1:1) 3 weeks after treatment administration. The date of birth and number of pups born per litter were recorded throughout a 7-month-long mating trial.

## Morphological and functional analysis of domestic cat sperm

Male kittens were anesthetized using the same anesthetic regimen as females. Semen collection was performed using a standardized electroejaculation protocol[62] at 3-, 6-, and 9-months post-injection. Briefly, electroejaculation was conducted using an electrostimulator and rectal probe (0.6 cm diameter, three longitudinal electrodes) (PT Electronics), delivering three separate sets of stimuli (20-30 stimuli/set) at increasing voltage (2–5 volts) with 5 min-long rest between sets. Recovered samples from each set were combined and assessed for seminal fluid volume, presence of sperm, sperm concentration, motility, and morphology[62]. At the 10-month post-treatment time point (1 year of age), males were anesthetized, semen collection was performed by urethral catheterization[35], and immediately followed by orchiectomy. Spermatozoa recovered from vasa deferentia and caudal epididymides were then combined with urethral catheter samples for analysis. Penile morphology (i.e., presence or absence of the preputial fold and penile spines) was assessed, and left (l) and right (r) testis length (L) and width (W) were measured with calipers at all four time points. The following formula was used to calculate the total testicular volume (cm³): $(L_l \times (W_l)^2 + L_r*(W_r)^2)*(\pi/6)$, which was compared to reference values[64]. Upon collection, testes and epididymides were fixed for 48 h in 10% formalin and transferred to 50% ethanol for storage. Tissue samples were processed in a Leica TP1020 automatic tissue processor (Leica Biosystems) and paraffin-embedded. Five-micron sections were stained with hematoxylin QS (Vector Laboratories) and eosin Y (MilliporeSigma) (H&E), coverslipped with CytoSeal 60 (Thermo Fisher Scientific), and examined under a Keyence BZ-X810 microscope (Keyence) for histological traits, presence of sperm, and seminiferous tubule measurements. The shortest and longest diameter of one round seminiferous tubule cross-section was measured on ten different fields of view per animal by a single blind observer using ImageJ version 1.53 m to obtain the average seminiferous tubule diameter.

To assess sperm function, semen collected from both treated males at 11 months of age was evaluated using in vitro fertilization (IVF) of in vitro matured (IVM) domestic cat oocytes[65]. Briefly, recovered semen from each male was diluted 1:1 in HEPES (MilliporeSigma)-buffered Feline-Optimized Culture Medium (FOCM)[66], concentrated by

centrifugation, and resuspended in FOCM-IVF to a concentration of $10 \times 10^6$ motile sperm/mL. Pre-equilibrated $20\,\mu L$ drops (under oil) received $5\,\mu L$ of diluted sperm ($2 \times 10^6$ motile sperm/mL), and incubated sperm drops were assessed 1- and 18-h later for motility traits. Oocytes retrieved 24 h earlier from the ovaries of spayed females and matured in FOCM-IVM medium (at 38.6 °C, 5% $CO_2$) were equilibrated in $47.5\,\mu L$ drops of FOCM-IVF medium and inseminated with $2.5\,\mu L$ of diluted sperm ($0.5 \times 10^6$ motile sperm/mL) for 18 h. Presumptive zygotes were then stripped of cumulus cells and loosely bound sperm by repeated pipetting through a narrow-bore pulled glass pipette, washed, and placed into fresh $50\,\mu L$ drops of FOCM-IVF for an additional 30 h of culture. At 48 h post-insemination, presumptive embryos were assessed (10-50X) for cleavage under a dissecting microscope and fixed in FOCM-Hepes containing 5% buffered formalin. Fluorescent staining was performed with $5\,\mu g$/mL Hoechst 33342 (MilliporeSigma) and was evaluated with an Axioscope 5 fluorescent microscope (ZEISS) to assess the maturation status of noncleaving oocytes and presence of blastomeric nuclei in embryos[65].

### Statistical analysis

Statistical analyses and graphic creation were conducted with GraphPad Prism version 9.3.1 (GraphPad Software). The number of biological replicates ($n$), the $P$-value, the specific statistical test performed, and if data are shown as SD or SEM are all included in the appropriate legend or main text. Differences were considered statistically significant when $P < 0.05$. A Student's $t$-test was used for comparisons between two groups of data. Datasets that were right-skewed and did not follow a normal distribution (i.e., vector genome copies in the blood, inhibin B, fecal hormone metabolites, and LH) were log-transformed before comparison. Apart from the low-dose cat that did not mate (*Bellatrix*) being removed for comparison of estrous behavior between controls and treated females, no data were excluded from the analyses.

Androgen metabolite baseline values were calculated for each male by averaging all data points before 5 months of age. Age at onset of puberty in males was defined as the first time that the average androgen metabolite levels of six consecutive fecal samples were two SD over baseline levels. Progestogen metabolite baseline values were calculated for each female using an iterative process, excluding all data points greater than the mean plus 3 SD, using the R statistical package hormLong version 1.0[67]. A luteal phase was defined as six or more consecutive fecal samples with a progestogen metabolites value greater than 3 SD over baseline.

### Reporting summary

Further information on research design is available in the Nature Portfolio Reporting Summary linked to this article.

## Data availability

The domestic cat (*Felis silvestris catus*) anti-Müllerian hormone (AMH) NCBI reference sequence mentioned in this work is XP_011286375.2 and can be found at https://www.ncbi.nlm.nih.gov/protein/XP_011286375.2. All data supporting the findings of this study are available within the paper and its Supplementary Information. A reporting summary is available as Supplementary Information file. Source data are provided with this paper.

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

## Acknowledgements
We deeply appreciate Scottie Fahey, Kim Vonder Meulen, Reagan Steele, Erin Russell, Haylie Kinman, Shelby Klotter, Sadie Phillips, and volunteers for their outstanding commitment and care of the domestic cats in the CREW colony. We are also grateful to Lynn Blattman, Mike Biere, Mike Camery, Jackie Dieckman, Linda McKinney, Ann Marchioni, and Mary Lou Zins for scoring and processing over 1500 h of mating trial footage. We thank Dr. Janine Brown and Stephen Paris at the Wildlife Endocrinology Research Laboratory (Smithsonian's National Zoo and Conservation Biology Institute) for serum LH analysis and Dr. Ashley Franklin for her assistance with statistical analysis of the hormone data. We thank Kirsten Coleman and Dr. Barry Byrne at the Powell Gene Therapy Center Toxicology Core (University of Florida) for quantification of vector shedding. This work was conducted with support from UM1TR004408 award through Harvard Catalyst | The Harvard Clinical and Translational Science Center (National Center for Advancing Translational Sciences, National Institutes of Health) and financial contributions from Harvard University and its affiliated academic healthcare centers. This work was supported with a grant (MG14-S06 to D.P., P.K.D., D.W., G.G., L.M.V., W.F.S.) from the Michelson Prize and Grants, a program of The Michelson Found Animals Foundation, which is supported by the generous contributions of Dr. Gary Michelson and Alya Michelson. L.M.V. and W.S. also gratefully acknowledge the Joanie Bernard Foundation for its financial support of this research. This work was also supported by the Department of Surgery of the Massachusetts General Hospital, the Huiying Foundation Patricia K. Donahoe, MD, Surgeon-Scientist Research Program: Mid-Career Catalyst Award, and the Sudna-Gar Fund (D.P.). P.G. was supported by a Postdoctoral Research Scholarship (B3X) from the *Fonds de recherche du Québec – Nature et Technologies* (*FQRNT*).

## Author contributions
P.K.D., L.R., T.J.C., W.F.S., L.M.V., and D.P. conceived and designed the experiments. D.W. and D.P. designed the transgene and vector. P.G., N.N., and N.S. performed the SAA, anti-AMH antibody, AMH, and inhibin B ELISAs. M.K. and D.P. performed mouse experiments. J.L.B., A.G.M., C.B., A.K.T., W.F.S., and L.M.V. carried out the health assessments, ultrasound examinations, male reproductive assessments, and fecal hormone metabolites measurements in cats. P.G. and L.M.V. performed the analysis of the breeding behavior. P.G., G.G., D.W., L.R., D.A.B., T.J.C., W.F.S., L.M.V., and D.P. supervised the work. The manuscript was drafted by P.G. and D.P. and edited by P.K.D., L.R., D.A.B., T.J.C., W.F.S., and L.M.V.

## Competing interests
The authors declare the following competing interests: Employees and advisors of The Michelson Found Animals Foundation were involved in the conceptualization and design of the study (L.R., D.A.B., and T.J.C.). P.K.D. and D.P. are co-inventors in the patent application #US 2024/0218038 A1, which relates to compositions and methods of reducing fertility in prepubescent non-human subjects by administering a composition comprising a vector comprising a nucleic acid encoding an anti-Müllerian hormone (AMH, also Mullerian inhibiting substance, MIS) protein. The remaining authors declare no competing interests.
