## [Transparent Peer Review file · Nature Communications]

Gene Therapy Delivery of Anti-Müllerian Hormone in Prepubertal Female Domestic Cats Induces Long-term Sterilization

Corresponding Author: Dr David Pepin

Version 0:

Reviewer comments:

Reviewer #1

(Remarks to the Author)

General comments

This paper is describing the successful sterilization of prepubertal female cats with a single injection of a adeno-associated viral vector delivering an anti-Müllerian hormone transgene, which is inducing high AMH levels during the almost 2 years of the study, without any notable effect on cat health or growth. The noteworthy results of this paper are the fact that this is the first time that the vaccination has been done in prepubertal female cats and male cats, with ensuing and probably lifelong sterility in female, but interestingly no sterility was present in male cats.

The authors have provided many detail on the methods used and have made an exhaustive list available of results (in the paper itself and in the supplemental data), over a longitudinal follow-up period of the cats and mice involved in the study, representing a large effort and a detailed study. The work is significant and will contribute to new developments in the field of non-surgical contraception in wildlife and domestic animals. As far as originality is concerned, the same group has previously demonstrated a similar approach in adult female cats (Vansandt, L. M. et al. Durable contraception in the female domestic cat using viral-vectored 814 delivery of a feline anti-Müllerian hormone transgene. *Nat. Commun.* 14, 3140 (2023)), but the data presented here are sufficiently different to be novel.

The methodology is sound and contains enough details for the work to be reproduced.

In conclusion, this is a generally well-written and well-executed study on a topic of interest to a broad scientific audience. I do have a few questions and suggestions for change, and also listed other suggestions for change below.

1. Line 73 : The claim that this type of vaccination could be of use for free roaming cat populations is somewhat exaggerated. Similar transgene vaccination to produce more AMH in feral (young) cats requires also trap- neuter-release (TNR) approach. It would be less invasive than surgical neutering but still requires i.m. injection. It is unclear whether the authors are aware of the classic paper by Blancou et al. (1986) on oral rabies vaccination in wild foxes

(<https://pubmed.ncbi.nlm.nih.gov/3736663/>), as well as other foundational studies by Pastoret on this topic. Using a similar oral approach for AMH may be hard to combine with a transgene vector and can be dangerous for inducing sterility in non-target carnivorous wildlife. But if the authors want to mention this type of vaccination as a tool to sterilize feral cats, they may have to consider alternative routes of administration as well here. They do mention an intranasal vaccin in the discussion.

2. Line 76 : The authors state : “Furthermore, in adult males of most vertebrate classes, AMH is produced by Sertoli cells and has been implicated in the regulation of testosterone production.^{21,22}” The most important cells implicated in testosterone production in the testis are the Leydig cells. To prevent confusion I suggest to mention Leydig cells here, since now it looks as if testosterone is produced by the Sertoli cells

3. In general, I found the order of the paragraphs in the discussion not very logical : (1) AMH levels increase and stay high (2)- no pregnancy in treated cats and then discussion on different modes of action with similar AMH vaccins ,(3) no ovulation and then discussion on anovulatory phenotype and reproductive hormones and so on. Why discussing first pregnancy before ovulation? I suggest to start with high AMH and how viral vectors act and then focus in the second paragraph on how high AMH may act on the ovary and prevent ovulation in the female cats as it was observed in the study and then continue towards mating and pregnancy. And to end with the males.

4. The authors repeat in the discussion many details that were already described in methods and results (from line 353 until 363). I would prefer to get more details on the presumed mechanism of actions. In line 363 – the authors speculate that, “given the efficient transduction of skeletal myocytes by AAV9 capsids, this increased production may be due to a combination of better transduction of young muscle cells and increases in muscle mass during puberty, which could

translate into a more durable sterilization efficacy and/or the potential for reducing the treatment dose. “ Before I read this sentence I was not aware that the skeletal muscles were produced the transgene (or do I understand it wrong). So it is not the granulosa cells in the ovary that are triggered and producing more AMH? Please explain this better. I would prefer to get more information on that topic here than the repeating previous 10 lines

5. I would like to read more speculation /explanation about the possible route of action of this vaccination and the possible effects of high AMH levels . Some studies (on male mice) have evaluated gene expression changes in Leydig cells as a results of high AMH administration. A recent study by the same authors (<https://doi.org/10.1073/pnas.241473412>) has described that AMH protects the ovarian reserve from chemotherapy (with description of pathways involved). There are probably many more studies that describe possible mechanism involved in AMH inhibitory effect on ovulation. One logical idea for consideration could be the fact that AMH is higher in prepuberal (non-ovulating) animals, is lower after puberty and is becoming very low in mammals in menopause (indicative of follicular depletion). Then again, AMH is high under pathological conditions like Sertoli cell tumor in dogs and Polycystic Ovarian Syndrome in women. Women with PCOS typically have a higher number of small antral follicles, which overproduce AMH. AMH levels in PCOS can be 2–3 times higher than normal and PCOS is also linked with anovulation. It may be interesting to look up some papers on these conditions for a more in-depth discussion

Specific comments

Line 740 Fluorescent staining was performed with 5 µg/mL Hoescht 33342 - spelling mistake should be Hoechst 33342

Line 1008 (Figure 1) the horizontal dashed line indicates the lowest AMH reading (0.54 µg/mL) ever measured in an AAV9-fcMISv2-treated sterile adult female domestic cat – Please also insert the reference here or is this from unpublished work?

Reviewer #2

(Remarks to the Author)

This manuscript is an interesting and very clear phenotypic analysis of the effects of AAV-based delivery of AMH on feline reproduction and some reproductive hormone parameters, when delivered to cats early in life, pre-puberty. The key results – that female but not male fertility is eliminated following AMH expression – are straightforward and convincing, even with a very modest sample size. The treatment also appears to be safe, as evidenced by weight gain, lack of evidence of systemic inflammation, tissue damage markers and injection site reactions. Breeding assays uncover some interesting differences from their earlier work on AAV-AMH treated adults, but the bottom line is the same: while breeding attempts do occur (more often when treated as juveniles than as adults), the females are infertile, even at low dose. Importantly, the AMH titer/vector genome is higher than observed in earlier work on adults.

This work derives particular significance as it follows several recent publications, also in Nature Communications, which demonstrated inhibition of progeny production when AMH was delivered to adult female cats, with these studies themselves following a first study of AAV-based AMH overexpression in female mice. The two earlier adult cat focused studies came to different conclusions regarding the mechanism of AMH action, pre- or post-fertilization. While the current manuscript does not resolve these differences (though the results are consistent with the authors previous argument for a pre-fertilization mechanism of action), it provides additional data that adds new wrinkles to our emerging understanding of the complexity of AMH effects on ovarian physiology.

Below my comments focus primarily on several points where I think the authors could expand on their discussion to clarify not only what are points of disagreement regarding mechanism of action, but what key experiments need to be carried out (notwithstanding qualms about isolating tissues and/or sacrificing animals at some point) to gain clarity.

With respect to the above (my suggestions) the authors do make clear, in lines 80 onwards, that their primary goal in this manuscript is to characterize phenotypic effects, not to solve mechanistic questions. Nonetheless, given the different mechanisms of action proposed, I still think it is important for the authors to address the topic.

Somewhere around line 66, or a bit later, it would be useful to first review prior results on the effects of AMH overexpression on female mouse and cat fertility. Not a deep dive, but just enough to get the reader oriented to the observations made arguing for both pre- and post-fertilization mechanisms of action, a topic returned to in the discussion.

As noted above, one interesting observation is that the titers of AMH observed/vector genome are generally higher than those observed following delivery of equivalent doses in adults. The basis for this is unclear but may be important in the context of other gene therapies where achieving high dose is critical. If there were other data in the literature that speaks to this kind of difference (perhaps from the muscular dystrophy literature), it would be useful to note it.

The authors limit themselves to non-invasive methods of analysis. Nonetheless, they are able to infer plausible mechanisms of action. In particular, several lines of evidence presented are consistent with induction of a hypergonadotropic hypogonadic state in which ovulation does not occur and cannot be induced by mating.

In thinking about all of this, and in addition the Stocker work and the earlier cat and mouse paper from these authors, it would be useful if the authors could, in the discussion, frame their thoughts a bit more with an eye to detailing for the readers exactly what experiments are needed to provide more definitive insights into how ovarian physiology and other hormonal axes are altered by AMH overexpression, even if the experiments needed are invasive and in some cases terminal for the animal (histology, blood analysis, RNAseq, etc). I appreciate that these authors may be limited in what they can do by funding-based restrictions on certain kinds of experiments. That said, inference is not the same as mechanistic demonstration, and it will help the reader a lot (and provide guidance to others interested in participating in the field) if the authors can draw out more explicitly hypotheses, predictions and experiments that will provide more definitive answers. The Stocker paper, for example, achieves a bit of a compromise in this regard by characterizing single ovaries from some of the

treated animals, thereby providing important information the current authors cannot obtain, while maintaining the AAV - treated cats for future study. Finally, if the authors have plans for various kinds of follow up characterization of these same animals that constrains their ability to do more invasive experiments, noting this at some point in the text would also give the reader some context for understanding the experimental path chosen. It can otherwise be confusing for the naive reader to understand why the authors "didn't just look" in ways that would otherwise commonly come to the mind of a student or practitioner of veterinary reproduction.

Bruce Hay

Reviewer #3

(Remarks to the Author)

Manuscript NCOMMS-25-35733: Gene Therapy Delivery of Anti-Müllerian Hormone in Prepubertal Female Domestic Cats Induces Long-term Sterilization, by Godin et al.

General comments:

This is a well-written manuscript that describes the impact of the administration of AMH via an AAV intra-muscular injection on the reproductive biology of kittens, including the exciting result that female kittens were rendered infertile as adults. The study design, data, and figures are entirely clear. I offer some minor edits for consideration, and I hesitate to request any additional work, b/c the investigators were very thorough. The authors indicated the cats, save for the breeding males, were made available for adoption (L508-509), and I wonder if the females were spayed pre-adoption and if their ovaries and uteri were available for interrogation? If these tissues were available for analysis, it be wonderful to now if there were any obvious differences between the controls and AAV-AMH-treated animals. For example, were tissue masses reduced in the AAV-AMH females and/or did their ovaries lack CLs on gross inspection? Data from the actual uteri would nicely complement the ultrasound findings that were reported.

- Recommend changing 'level(s)' to 'concentration(s)' throughout in instances when level(s) referred to the concentrations of hormones.
- Recommend consulting with a statistician as to which statistical tests are allowed when the sample size for a group is less than 3, in particular, stats analyses of the outcomes that involved the control female cats, for which the sample size was 2. And here are two additional stats related questions:
 - o L977, Table 1. Should the Fisher's exact test for the number of pregnancies been two-sided rather than one-sided?
 - o Figure 2f. The sample sizes for males were one animal per group; can stats be performed on sample sizes of 1?

Specific Comments (by line):

L249-250. Because male fertility wasn't tested, it would be more accurate to state "...likely does not prevent male fertility".

L.262 and L.263. Between these lines, it would be a better place to mention that domestic cats are considered to be induced ovulators, but several studies have found that a proportion of females will ovulate spontaneously. Better here than waiting until L334.

L263. Insert 'as metabolites' between excreted and primarily.

L.288 Insert 'spontaneous' between inhibit and ovulation.

L327. Change 'estrus' to 'estrous'.

L508-509. Perhaps mention what happened to the ovaries and uteri if cats were spayed before being adopted out.

L568. Because samples were collected into EDTA tubes, the non-cellular fraction would be plasma rather than serum. Please change 'serum' to 'plasma' in L573 and elsewhere. Also, double check that all analytical methods used on these samples were valid for EDTA plasma.

L680. Because domestic cats are seasonal breeders, did all females undergo the four-month-long breeding trials at the same time of year? Even though photoperiod was held at 14 h light per day, there could be a circannual rhythm to feline breeding, and depending on the time of year, some cats could potentially discontinue estrous cycling.

L691. Change 'estrus' to 'estrous'.

Supplementary Figure 4. Within the legend, it would be worth bringing to the reader's attention that the Y-axis range for the AMH-treated females (0.00-50 ug/mL) is markedly greater than that for the control females (0.00-0.10 ug/mL).

Version 1:

Reviewer comments:

Reviewer #1

(Remarks to the Author)

The manuscript has been revised according to my comments and that of the other reviewers. The points I raised were sufficiently explained.

Reviewer #2

(Remarks to the Author)

The authors have address my comments, and (I think) the comments of the other reviewers.

Bruce Hay

Reviewer #3

(Remarks to the Author)

I am satisfied with the authors' response to my reviews.

RESPONSE TO REFEREES

NCOMMS-25-35733: Gene Therapy Delivery of Anti-Müllerian Hormone in Prepubertal Female Domestic Cats Induces Long-term Sterilization, by Godin et al.

Thank you for your consideration of our manuscript. We have considered the reviewers' comments very carefully and have addressed all suggestions with responses and modifications and thank them for their constructive feedback.

Importantly, we have added more information regarding how AAV therapies are able to induce the sustained expression of a transgene and discussed experiments needed to better understand how AMH overexpression can impact ovarian physiology.

The authors' point-by-point responses to the reviewers' comments are **bolded**.

REVIEWER COMMENTS

Reviewer #1 (Remarks to the Author):

General comments

This paper is describing the successful sterilization of prepubertal female cats with a single injection of a adeno-associated viral vector delivering an anti-Müllerian hormone transgene, which is inducing high AMH levels during the almost 2 years of the study, without any notable effect on cat health or growth. The noteworthy results of this paper are the fact that this is the first time that the vaccination has been done in prepubertal female cats and male cats, with ensuing and probably lifelong sterility in female, but interestingly no sterility was present in male cats.

The authors have provided many detail on the methods used and have made an exhaustive list available of results (in the paper itself and in the supplemental data), over a longitudinal follow-up period of the cats and mice involved in the study, representing a large effort and a detailed study. The work is significant and will contribute to new developments in the field of non-surgical contraception in wildlife and domestic animals. As far as originality is concerned, the same group has previously demonstrated a similar approach in adult female cats (Vansandt, L. M. et al. Durable contraception in the female domestic cat using viral-vectored 814 delivery of a feline anti-Müllerian hormone transgene. Nat. Commun. 14, 3140 (2023)), but the data presented here are sufficiently different to be novel.

The methodology is sound and contains enough details for the work to be reproduced.

In conclusion, this is a generally well-written and well-executed study on a topic of interest to a broad scientific audience. I do have a few questions and suggestions for change, and also listed other suggestions for change below.

We thank the reviewer for acknowledging the substantial effort and level of detail invested in this study, and we sincerely appreciate their kind words regarding the quality of the writing and execution. We also appreciate their mention of the significance of this work to the field of non-surgical contraception in wildlife and domestic animals.

1. Line 73 : The claim that this type of vaccination could be of use for free roaming cat populations is somewhat exaggerated. Similar transgene vaccination to produce more AMH in

feral (young) cats requires also trap- neuter-release (TNR) approach. It would be less invasive than surgical neutering but still requires i.m. injection. It is unclear whether the authors are aware of the classic paper by Blancou et al. (1986) on oral rabies vaccination in wild foxes (<https://pubmed.ncbi.nlm.nih.gov/3736663/>), as well as other foundational studies by Pastoret on this topic. Using a similar oral approach for AMH may be hard to combine with a transgene vector and can be dangerous for inducing sterility in non-target carnivorous wildlife. But if the authors want to mention this type of vaccination as a tool to sterilize feral cats, they may have to consider alternative routes of administration as well here. They do mention a intranasal vaccin in the discussion.

We thank the reviewer for raising this important point. Indeed, a limitation of the proposed strategy is that it requires the capture of the animals to be individually injected. While this step mirrors some of the logistical demands of trap–neuter–release (TNR) programs, we note that an IM injection is considerably faster and less resource-intensive than ovariohysterectomy, requiring fewer personnel hours per animal and posing no risk of surgical complications or post-operative pain. Capturing the domestic cats also allows the marking of treated individuals (e.g., ear notching, tattooing), which is essential for long-term population management and for preventing unnecessary recapture.

Regarding the reviewer’s concern that our claim may be somewhat exaggerated, we respectfully clarify our position. Line 73 of the manuscript (now Lines 75-79) reads “...would be a potent tool in managing free-roaming cat populations.” This statement is intended to convey potential utility, not to suggest that AAV-based contraception will replace surgical sterilization or serve as a stand-alone solution to feline overpopulation. Rather, we view this approach as a complementary tool, particularly valuable in settings where access to medical supplies, anesthesia, analgesia, and trained veterinary personnel is limited.

With regards to these two elements, in the Discussion section (starting on Line 478), we replaced:

- **“In a field setting where medical history is often unknown and postoperative care is limited, the use of a non-surgical sterilant would present many safety advantages.”**

With the following sentences:

- **“In a field setting, this new approach would also require the capture of cats. While this mirrors a logistical requirement of TNR, administering an intramuscular product is considerably faster and less resource-intensive than surgical spay, requiring fewer personnel hours per animal. Under conditions where medical history is often unknown and access to medical supplies and trained veterinary personnel is limited, a non-surgical sterilant used as a complementary tool to TNR could prove to be highly valuable and offer many safety advantages.”**

We also thank the reviewer for highlighting the parallels with vaccination strategies. While it is possible to use AAVs for the delivery of an antigen with the goal of eliciting an adaptive immune response resulting in the production of antibodies and immunological memory, we wish to clarify that the use of AAVs in our context represents a fundamentally different approach from vaccination. The AAV9-fcMISv2 therapy works by delivering the fcMISv2 transgene directly into the cells of the treated cat, which in turn secrete the transgene product (AMH) into the blood circulation without triggering the immune system. The

treated cat muscle cells essentially become a biological factory producing the therapeutic AMH protein (more on the mechanism of AAV-based therapies in response to Reviewer1 Point#4).

We fully agree with the reviewer that the oral administration would be an easier method, especially for remote or large populations of free-roaming cats or for adapting this strategy for the control of smaller species such as rodents. However, the production cost of AAV-based therapies makes widespread environmental dissemination via bait impractical. There is currently limited data on the stability of AAV particles in outdoor conditions. However, in laboratory conditions, UV exposure was shown to significantly impact vector activity^{1,2}. Stability of recombinant AAVs can also be compromised in low-pH conditions³, effective oral delivery would likely require formulation in encapsulated forms to protect the viral particles and enable release in the lower GI tract—a challenge that has yet to be addressed in this context. We also appreciate the reviewer’s point regarding the potential risk of sterilizing non-target carnivores, particularly felids, if any AAV-based sterilant were to be deployed as bait.

Lastly, while the intranasal delivery of AAVs has been explored in contexts where transfection of airway or lung tissues is desirable, we do not believe it is a suitable route for the AAV9-fcMISv2 therapy since it requires broad tissue distribution.

- ¹Gruntman, A.M., Su, L., Su, Q., Gao, G., Mueller, C. and Flotte, T.R., 2015. Stability and compatibility of recombinant adeno-associated virus under conditions commonly encountered in human gene therapy trials. *Human gene therapy methods*, 26(2), pp.71-76.
- ²Tomono, T., Hirai, Y., Chono, H., Mineno, J., Ishii, A., Onodera, M., Tamaoka, A. and Okada, T., 2019. Infectivity assessment of recombinant adeno-associated virus and wild-type adeno-associated virus exposed to various diluents and environmental conditions. *Human Gene Therapy Methods*, 30(4), pp.137-143.
- ³Lengler, J., Gavrilu, M., Brandis, J., Palavra, K., Dieringer, F., Unterthurner, S., Fuchsberger, F., Kraus, B. and Bort, J.A.H., 2024. Crucial aspects for maintaining rAAV stability. *Scientific Reports*, 14(1), p.27685.

2. Line 76 : The authors state : “Furthermore, in adult males of most vertebrate classes, AMH is produced by Sertoli cells and has been implicated in the regulation of testosterone production.21.22 ” The most important cells implicated in testosterone production in the testis are the Leydig cells. To prevent confusion I suggest to mention Leydig cells here, since now it looks as if testosterone is produced by the Sertoli cells

Thank you for the suggestion. The sentence (now Line 81) now reads: “Furthermore, in adult males of most vertebrate classes, AMH is produced by Sertoli cells and has been implicated in the regulation of testosterone production by Leydig cells.”.

3. In general, I found the order of the paragraphs in the discussion not very logical : (1) AMH levels increase and stay high (2)- no pregnancy in treated cats and then discussion on different modes of action with similar AMH vaccins ,(3) no ovulation and then discussion on anovulatory phenotype and reproductive hormones and so on. Why discussing first pregnancy before ovulation? I suggest to start with high AMH and how viral vectors act and then focus in the second paragraph on how high AMH may act on the ovary and prevent ovulation in the female

cats as it was observed in the study and then continue towards mating and pregnancy. And to end with the males.

We thank the reviewer for this suggestion. The modifications we made to the first paragraph of the Discussion in response to Reviewer1-Point#4 simplify the discussion of AMH levels and also adds to the explanation on how AAV-based therapies work.

We decided to discuss the lack of pregnancy in our study and how these results compare with those presented in Stocker et al., 2025 before discussing the anovulatory phenotype because our study was designed to evaluate the efficacy of the AAV9-fcMISv2 sterilant rather than characterizing its mechanism of action. In that sense, we discuss the lack of pregnancy that we observed before because it is more directly tied to our study goal. We acknowledge that if this study were to be more mechanistically-oriented, we would certainly discuss these elements in the reverse order.

4. The authors repeat in the discussion many details that were already described in methods and results (from line 353 until 363). I would prefer to get more details on the presumed mechanism of actions. In line 363 – the authors speculate that, “given the efficient transduction of skeletal myocytes by AAV9 capsids, this increased production may be due to a combination of better transduction of young muscle cells and increases in muscle mass during puberty, which could translate into a more durable sterilization efficacy and/or the potential for reducing the treatment dose. “ Before I read this sentence I was not aware that the skeletal muscles were produced the transgene (or do I understand it wrong). So it is not the granulosa cells in the ovary that are triggered and producing more AMH? Please explain this better. I would prefer to get more information on that topic here than the repeating previous 10 lines

Thank you for highlighting the section between Lines 353-363 (now Lines 367-376). We deleted one sentence (starting on Line 375) to shorten this paragraph. Although it repeats some previously mentioned results and methods, we feel that the rest of this section is useful as it opens the Discussion section.

We thank the reviewer for pointing out the need for clarifications on the mechanism of action of AAVs when used in such therapies.

When administered systemically, AAVs disseminate to a wide variety of tissues^{1,2}. In our study, the exact nature of the cells transfected by AAV9-fcMISv2 is not the primary concern as long as it allows sufficiently high and sustained AMH levels to inhibit ovarian activity for several years. However, to achieve AMH concentrations in the $\mu\text{g}/\text{mL}$ range for multiple years following a single injection, a high proportion of the transfected cells must be long-lived and terminally differentiated. Since AAV-delivered transgene copies are not replicated during cell division (the transgene does not integrate the genome, it stays extrachromosomal), extensive tissue remodeling (via cell proliferation and death) results in the eventual dilution and loss of transgene copies. For example, epithelial cells are poor targets for lasting expression in AAV-based gene therapies because of their fast turnover.

The skeletal muscle compartment is ideal for long-term transgene expression because, in absence of muscle injury, adult skeletal myocytes do not divide and can thus constantly produce the therapeutic protein. We used the AAV9 serotype because it has been used efficiently in the past and has a good transfection efficiency of the skeletal muscle cells^{2,3}.

Thus, the goal of our AAV9-fcMISv2 therapy is not for the AMH transgene to be transduced into the granulosa cells but rather we wish to transfect long-lived, differentiated cells of the cat (such as skeletal muscle cells) so that the AMH protein is secreted into the circulation and reaches the ovary to inhibit its activity.

Upon suggestion of the reviewer, we further clarified the mechanism of this AAV-based gene therapy, with the following sentences to the revised manuscript:

- Line 61: “To date, all AAV-based vectored contraception approaches have used rAAVs to convert transduced tissues into biofactories that continuously secrete the contraceptive protein into the bloodstream, allowing its delivery to the gonads.”
 - Line 70: “AAV9 has a high tropism for skeletal muscle cells, which are long-lived and terminally differentiated, making it an ideal serotype for achieving a sustained expression of the therapeutic transgene.”
- ¹Zincarelli, C., Soltys, S., Rengo, G. and Rabinowitz, J.E., 2008. Analysis of AAV serotypes 1–9 mediated gene expression and tropism in mice after systemic injection. *Molecular therapy*, 16(6), pp.1073-1080.
- ²Walkey, C.J., Snow, K.J., Bulcha, J., Cox, A.R., Martinez, A.E., Ljungberg, M.C., Lanza, D.G., De Giorgi, M., Chuecos, M.A., Alves-Bezerra, M. and Suarez, C.F., 2025. A Comprehensive Atlas of AAV Tropism in the mouse. *Molecular Therapy*.
- ³Wang, D., Tai, P. W. & Gao, G. Adeno-associated virus vector as a platform for gene therapy delivery. *Nat. Rev. Drug Discov.* 18, 358-378 (2019).

5. I would like to read more speculation /explanation about the possible route of action of this vaccination and the possible effects of high AMH levels . Some studies (on male mice) have evaluated gene expression changes in Leydig cells as a results of high AMH administration. A recent study by the same authors (<https://doi.org/10.1073/pnas.241473412>) has described that AMH protects the ovarian reserve from chemotherapy (with description of pathways involved). There are probably many more studies that describe possible mechanism involved in AMH inhibitory effect on ovulation. One logical idea for consideration could be the fact that AMH is higher in prepuberal (non-ovulating) animals, is lower after puberty and is becoming very low in mammals in menopause (indicative of follicular depletion). Then again, AMH is high under pathological conditions like Sertoli cell tumor in dogs and Polycystic Ovarian Syndrome in women. Women with PCOS typically have a higher number of small antral follicles, which overproduce AMH. AMH levels in PCOS can be 2–3 times higher than normal and PCOS is also linked with anovulation. It may be interesting to look up some papers on these conditions for a more in-depth discussion

The current study was designed to evaluate the safety and sterilizing efficacy of AAV9-fcMISv2. While the hormonal data gathered allows us to speculate the anovulatory phenotype secondary to elevated AMH, the unavailability of ovarian tissue prevented the exploration of the cellular and molecular determinants of this.

As referenced on Line 209, we are aware of the reports on the AMH impact on *Cyp17a1* expression and testosterone production of Leydig cells. But without any gene expression data in cat ovaries, we can't yet speculate on if it is also the case in females.

In the Discussion paragraph that relates to the proposed mechanism of AMH action, we reference our studies (Meinsohn et al., 2021 (ref#19); Nguyen et al., 2025 (ref#53) – Line

425) that reported molecular signatures of AMH treatment on granulosa cells. Notably, we showed that AMH inhibited the proliferation of multiple type of ovarian cells, that it uncoupled the maturation of germ cells and granulosa cells and that it resulted in a lower number of growing preantral follicles. These could be involved in the mechanism of anovulation reported here. However, we feel that we haven't generated the necessary data to speculate further on either of these options.

We added the following sentence on Lines 425-427: "We speculate that a similar block in follicular development may occur in treated cats but is generally masked by endocrine homeostatic processes."

Most of the circulating AMH in healthy females is produced by preantral and small antral follicles. This explains why peri-pubertal girls have higher AMH levels than adult women, and why post-menopausal women with follicle-depleted ovaries have very low AMH. However, prepubertal females are non-ovulating mostly because of the immaturity of their Hypothalamic-Pituitary-Gonadal (HPG) axis¹, rather than AMH-related causes.

As mentioned by the reviewer, pathological conditions can result in increased secretion of AMH by the gonads. While the etiology of anovulation in Polycystic Ovary Syndrome (PCOS) is complex, the elevated AMH is thought to be mostly secondary to an imbalance in gonadotropin production and secretion leading to an accumulation of anovulatory antral follicles, and elevated androgen production. Our in vivo data generated to date in various animal models (mice, rats, cats) suggest that ectopic supraphysiological AMH does not cause PCOS, since its main canonical features (elevated androgens, antral follicle cyst accumulation) are not recapitulated. Although we concede that there may be a similarity in the anovulatory phenotype, we think that drawing parallels between our sterilizing approach and PCOS might confuse the reader into thinking that our treatment could recapitulate other hallmarks of PCOS.

- ¹Spaziani, M., Tarantino, C., Tahani, N., Gianfrilli, D., Sbardella, E., Lenzi, A. and Radicioni, A.F., 2021. Hypothalamo-Pituitary axis and puberty. *Molecular and cellular endocrinology*, 520, p.111094.

Specific comments

Line 740 Fluorescent staining was performed with 5 µg/mL Hoescht 33342 - spelling mistake should be Hoechst 33342

Changed as suggested.

Line 1008 (Figure 1) the horizontal dashed line indicates the lowest AMH reading (0.54 µg/mL) ever measured in an AAV9-fcMISv2-treated sterile adult female domestic cat – Please also insert the reference here or is this from unpublished work?

This is indeed from previously published work. The specific reference was added.

Reviewer #2 (Remarks to the Author):

This manuscript is an interesting and very clear phenotypic analysis of the effects of AAV-based delivery of AMH on feline reproduction and some reproductive hormone parameters, when

delivered to cats early in life, pre-puberty. The key results – that female but not male fertility is eliminated following AMH expression – are straightforward and convincing, even with a very modest sample size. The treatment also appears to be safe, as evidenced by weight gain, lack of evidence of systemic inflammation, tissue damage markers and injection site reactions. Breeding assays uncover some interesting differences from their earlier work on AAV-AMH treated adults, but the bottom line is the same: while breeding attempts do occur (more often when treated as juveniles than as adults), the females are infertile, even at low dose. Importantly, the AMH titer/vector genome is higher than observed in earlier work on adults.

This work derives particular significance as it follows several recent publications, also in Nature Communications, which demonstrated inhibition of progeny production when AMH was delivered to adult female cats, with these studies themselves following a first study of AAV-based AMH overexpression in female mice. The two earlier adult cat focused studies came to different conclusions regarding the mechanism of AMH action, pre- or post-fertilization. While the current manuscript does not resolve these differences (though the results are consistent with the authors previous argument for a pre-fertilization mechanism of action), it provides additional data that adds new wrinkles to our emerging understanding of the complexity of AMH effects on ovarian physiology.

We thank the reviewer for acknowledging the significance of this work and its contribution to the growing body of knowledge of how elevated AMH affects ovarian physiology.

Below my comments focus primarily on several points where I think the authors could expand on their discussion to clarify not only what are points of disagreement regarding mechanism of action, but what key experiments need to be carried out (notwithstanding qualms about isolating tissues and/or sacrificing animals at some point) to gain clarity.

With respect to the above (my suggestions) the authors do make clear, in lines 80 onwards, that their primary goal in this manuscript is to characterize phenotypic effects, not to solve mechanistic questions. Nonetheless, given the different mechanisms of action proposed, I still think it is important for the authors to address the topic.

Somewhere around line 66, or a bit later, it would be useful to first review prior results on the effects of AMH overexpression on female mouse and cat fertility. Not a deep dive, but just enough to get the reader oriented to the observations made arguing for both pre- and post-fertilization mechanisms of action, a topic returned to in the discussion.

Thank you for the suggestion, we included a mention of the Stocker et al. 2025 study in the introduction section (Line 75).

As noted above, one interesting observation is that the titers of AMH observed/vector genome are generally higher than those observed following delivery of equivalent doses in adults. The basis for this is unclear but may be important in the context of other gene therapies where achieving high dose is critical. If there were other data in the literature that speaks to this kind of difference (perhaps from the muscular dystrophy literature), it would be useful to note it.

We thank the reviewer for suggesting reviewing the literature on the impact of age on the ability of AAVs to transduce muscle cells efficiently.

Most of the studies that looked at the impact of age on muscle tissue transduction compared neonatal and adult animals. We couldn't find studies that specifically compared prepubescent (21-42 days old mice, 2-4 months old cats or dogs) with adult individuals.

A study by Lovric et al.¹ reports that cardiomyocytes and skeletal muscle cells are more permissive to AAV transduction once they are terminally differentiated. They showed that the administration of an equivalent dose of rAAVs results in a higher expression in the cardiac muscle in 21 days old than in 7 days old mice. They didn't compare with adult mice, but their results suggest that high permissivity of muscle cells to AAVs is already attained at 21 days of age.

Bostick et al.² report an efficient and equivalent transduction of skeletal muscles following the IV delivery of rAAVs in newborn and adult mice.

Data from the muscular dystrophy literature suggest that the expression of an AAV-delivered transgene in skeletal muscle cells is lower in young adult than in neonatal mice³, but that this difference might be secondary to cellular changes due to disease status.

To highlight this gap of knowledge in the gene therapy literature, we added the following sentence to the end of the first paragraph of Discussion (Lines 381-384): "To our knowledge, this is the first report of increased transgenic protein expression following the intramuscular administration of an AAV-based therapy to prepubertal animals compared to fully developed adults."

- ¹Lovric, J., Mano, M., Zentilin, L., Eulalio, A., Zacchigna, S. and Giacca, M., 2012. Terminal differentiation of cardiac and skeletal myocytes induces permissivity to AAV transduction by relieving inhibition imposed by DNA damage response proteins. *Molecular Therapy*, 20(11), pp.2087-2097.
- ²Bostick, B., Ghosh, A., Yue, Y., Long, C. and Duan, D., 2007. Systemic AAV-9 transduction in mice is influenced by animal age but not by the route of administration. *Gene therapy*, 14(22), pp.1605-1609.
- ³Ghosh, A., Yue, Y., Shin, J.H. and Duan, D., 2009. Systemic Trans-splicing adeno-associated viral delivery efficiently transduces the heart of adult mdx mouse, a model for duchenne muscular dystrophy. *Human gene therapy*, 20(11), pp.1319-1328.

The authors limit themselves to non-invasive methods of analysis. Nonetheless, they are able to infer plausible mechanisms of action. In particular, several lines of evidence presented are consistent with induction of a hypergonadotropic hypogonadic state in which ovulation does not occur and cannot be induced by mating.

In thinking about all of this, and in addition the Stocker work and the earlier cat and mouse paper from these authors, it would be useful if the authors could, in the discussion, frame their thoughts a bit more with an eye to detailing for the readers exactly what experiments are needed to provide more definitive insights into how ovarian physiology and other hormonal axes are altered by AMH overexpression, even if the experiments needed are invasive and in some cases terminal for the animal (histology, blood analysis, RNAseq, etc). I appreciate that these authors may be limited in what they can do by funding-based restrictions on certain kinds of experiments. That said, inference is not the same as mechanistic demonstration, and it will help the reader a lot (and provide guidance to others interested in participating in the field) if the

authors can draw out more explicitly hypotheses, predictions and experiments that will provide more definitive answers. The Stocker paper, for example, achieves a bit of a compromise in this regard by characterizing single ovaries from some of the treated animals, thereby providing important information the current authors cannot obtain, while maintaining the AAV -treated cats for future study. Finally, if the authors have plans for various kinds of follow up characterization of these same animals that constrains their ability to do more invasive experiments, noting this at some point in the text would also give the reader some context for understanding the experimental path chosen. It can otherwise be confusing for the naive reader to understand why the authors "didn't just look" in ways that would otherwise commonly come to the mind of a student or practitioner of veterinary reproduction.

Bruce Hay

We thank the reviewer for this suggestion and for acknowledging limitations that can occur in performing this type of study. Female cats were adopted intact to see if they would show altered behavior when living in a home environment. We agree that it would be a good addition to the Discussion section to detail to the reader what experiments are needed to better understand how AMH overexpression can impact ovarian physiology.

We added the following to the third paragraph of the Discussion (Lines 427-433): “Future studies, such as comprehensive histological assessments of the ovaries of AAV9-fcMISv2-treated females with follicle counts, along with single-cell transcriptomics, could provide a more detailed characterization of the impact of sustained supraphysiological AMH on domestic cat ovarian physiology and help elucidate the mechanism of anovulation. However, female domestic cats of this study were adopted intact to investigate their behavior in a home environment, precluding us from conducting such analyses.”

Reviewer #3 (Remarks to the Author):

Manuscript NCOMMS-25-35733: Gene Therapy Delivery of Anti-Müllerian Hormone in Prepubertal Female Domestic Cats Induces Long-term Sterilization, by Godin et al.

General comments:

This is a well-written manuscript that describes the impact of the administration of AMH via an AAV intra-muscular injection on the reproductive biology of kittens, including the exciting result that female kittens were rendered infertile as adults. The study design, data, and figures are entirely clear. I offer some minor edits for consideration, and I hesitate to request any additional work, b/c the investigators were very thorough. The authors indicated the cats, save for the breeding males, were made available for adoption (L508-509), and I wonder if the females were spayed pre-adoption and if their ovaries and uteri were available for interrogation? If these tissues were available for analysis, it be wonderful to now if there were any obvious differences between the controls and AAV-AMH-treated animals. For example, were tissue masses reduced in the AAV-AMH females and/or did their ovaries lack CLs on gross inspection? Data from the actual uteri would nicely complement the ultrasound findings that were reported.

We thank the reviewer for recognizing the quality of the writing, clarity of the results and figures and the thoroughness of our study. The female domestic cats of this study were adopted intact to follow their behavior in a home environment. We plan to analyze reproductive organs of AAV9-fcMISv2 females in a future study.

• *Recommend changing ‘level(s)’ to ‘concentration(s)’ throughout in instances when level(s) referred to the concentrations of hormones.*

Thank you for the recommendation. We changed every occurrence of “levels” to “concentrations” where it referred to the concentrations of a hormone measured in the serum.

• *Recommend consulting with a statistician as to which statistical tests are allowed when the sample size for a group is less than 3, in particular, stats analyses of the outcomes that involved the control female cats, for which the sample size was 2.*

And here are two additional stats related questions:

o L977, Table 1. Should the Fisher’s exact test for the number of pregnancies been two-sided rather than one-sided?

In our understanding, a one-sided statistical test can be used if there is a strong assumption (before generating the data) about the direction of an effect and if the consequences of missing an effect in the other direction are negligible. Importantly, we did not decide to apply a one-sided Fisher’s exact test following the non-significant results of a two-sided test.

Based on the previous results of lower rate of pregnancy in adult female cats treated with AAV9-fcMISv2 (Vansandt et al., 2023), our prior assumption was that the number of pregnancies in the treated group would be lower than in the control group. The one-sided Fisher’s exact test used in Table 1 tests if the values of pregnancy in the treatment group are lower or equal to the control group. In doing so, we fail to test for the possibility that the treatment increases pregnancy rate in treated females. If the association went the other way (higher rate of pregnancies in treated females) using our one-sided Fisher’s exact test, we would have concluded that the difference is due to chance and therefore not statistically significant.

We are confident to use a one-sided Fisher’s exact test in this instance because our conclusions on both groups being equal and on the treated group having higher rates of pregnancies would be the same (ie. we would stop pursuing the AAV9-fcMISv2 treatment as a potential sterilant).

o Figure 2f. The sample sizes for males were one animal per group; can stats be performed on sample sizes of 1?

Thank you for pointing this out to us. We removed the statistical comparison from Figure 2F.

Specific Comments (by line):

L249-250. Because male fertility wasn’t tested, it would be more accurate to state “...likely does not prevent male fertility”.

Changed as suggested.

L.262 and L.263. Between these lines, it would be a better place to mention that domestic cats

are considered to be induced ovulators, but several studies have found that a proportion of females will ovulate spontaneously. Better here than waiting until L334.

Thank you for the suggestion. To resolve this, we deleted the sentence on Line 334 (now 348) and we added the following two sentences on Lines 274-276: “The female domestic cat is classified as an induced ovulator, with copulation serving as the canonical stimulus for ovulation. In absence of copulation, certain females can also spontaneously ovulate.”

L263. Insert ‘as metabolites’ between excreted and primarily.

Changed as suggested.

L.288 Insert ‘spontaneous’ between inhibit and ovulation.

Thanks for the suggestion. We kept the original sentence because the use of “ovulation” here refers to both breeding-induced and spontaneous ovulations. Although we mention in the previous sentence (now Lines 295-298) that spontaneous ovulations were never observed in treated females and was observed once in a control, the conclusion sentence aims to englobe also the results of lower progesterone described on Lines 285-295. The absence of peak progesterone metabolite readings in treated females (even during the mating trials) suggest that supraphysiological AMH inhibits both types of ovulations.

L327. Change ‘estrus’ to ‘estrous’.

Changed as suggested.

L508-509. Perhaps mention what happened to the ovaries and uteri if cats were spayed before being adopted out.

Female cats were adopted intact, we added a sentence (Line 431) to inform the reader.

L568. Because samples were collected into EDTA tubes, the non-cellular fraction would be plasma rather than serum. Please change ‘serum’ to ‘plasma’ in L573 and elsewhere. Also, double check that all analytical methods used on these samples were valid for EDTA plasma.

Only blood samples collected to assess viral persistence in circulation were collected in EDTA tubes. Blood samples collected for all other purposes were left to clot in sample separator tubes and then centrifuged to isolate the serum. We added “in sample separator tubes (Avantik)” on line 601 to make it more clear for the reader. We also verified the whole manuscript to find instances where “plasma” should be used instead of the word “serum”.

L680. Because domestic cats are seasonal breeders, did all females undergo the four-month-long breeding trials at the same time of year? Even though photoperiod was held at 14 h light per day, there could be a circannual rhythm to feline breeding, and depending on the time of year, some cats could potentially discontinue estrous cycling.

We thank the reviewer for this thoughtful comment. All females underwent the four-month mating trial during the same time of year. In the “Mating trials” section (starting on Line 712) of the Methods, we refer to Supplementary Table 1 which indicates (second-to-last column) that the trial began on 2022-03-24 for all females. To make this even clearer for the

reader, we added the word “single” in Lines 320 (Results section), 713 (Methods) and 1,022 (Legend of Figure 1) to read “...a single four-month-long mating trial...”.

L691. Change ‘estrus’ to ‘estrous’.

Changed as suggested.

Supplementary Figure 4. Within the legend, it would be worth bringing to the reader’s attention that the Y-axis range for the AMH-treated females (0.00-50 ug/mL) is markedly greater than that for the control females (0.00-0.10 ug/mL).

We added a sentence in the legend of Supp. Fig. 4 that reads: “Note that the y-axis scale range differs between control (0.0-0.1) and treated (0-50) females.”